# InternScenes: A Large-scale Simulatable Indoor Scene Dataset with Realistic Layouts

**Weipeng Zhong**[1,2,*] **Peizhou Cao**[1,3*], **Yichen Jin**[4], **Li Luo**[1,5], **Wenzhe Cai**[1], **Jingli Lin**[1,2],
**Hanqing Wang**[1], **Zhaoyang Lyu**[1], **Tai Wang**[1], **Bo Dai**[5], **Xudong Xu**[1,†] **Jiangmiao Pang**[1,†]

[1]Shanghai AI Laboratory      [2]Shanghai Jiao Tong University
[3]Beihang University      [4]Peking University      [5]The University of Hong Kong

## Abstract

The advancement of Embodied AI heavily relies on large-scale, simulatable 3D scene datasets characterized by scene diversity and realistic layouts. However, existing datasets typically suffer from limitations in data scale or diversity, sanitized layouts lacking small items, and severe object collisions. To address these shortcomings, we introduce **InternScenes**, a novel large-scale simulatable indoor scene dataset comprising approximately 40,000 diverse scenes by integrating three disparate scene sources, *i.e.*, real-world scans, procedurally generated scenes, and designer-created scenes, including 1.96M 3D objects and covering 15 common scene types and 288 object classes. We particularly preserve massive small items in the scenes, resulting in realistic and complex layouts with an average of 41.5 objects per region. Our comprehensive data processing pipeline ensures simulatability by creating real-to-sim replicas for real-world scans, enhances interactivity by incorporating interactive objects into these scenes, and resolves object collisions by physical simulations. We demonstrate the value of InternScenes with two benchmark applications: scene layout generation and point-goal navigation. Both show the new challenges posed by the complex and realistic layouts. More importantly, InternScenes paves the way for scaling up the model training for both tasks, making the generation and navigation in such complex scenes possible. We commit to open-sourcing the data and benchmarks to benefit the whole community.

## 1 Introduction

In the realm of embodied intelligence, 3D scenes [12, 25, 10] serve as the basis of simulation environments and become increasingly essential for agents to acquire a wide range of skills [4, 18], thereby significantly facilitating the advancement of Embodied AI. To encourage agents to learn more diverse skills and robustly adapt to various application scenarios, the whole community warrants a large-scale 3D dataset characterized by diverse and realistic layouts. While the dataset diversity refers to the richness and variety of scenes, encompassing a multitude of 3D object types, a realistic layout entails complex relationships between objects and a large number of objects within regions, especially small items. More importantly, the inclusion of various interactive objects in the scenes is crucial to support the learning of diverse agent skills.

Unfortunately, existing datasets fall short of meeting the aforementioned requirements and can be broadly categorized into three groups. 1) Real-world scanned scenes [8, 37, 5] boast realistic layouts and originate from a vast and diverse range of sources. However, these scanned data are typically represented as point clouds, having incomplete or inaccurate geometry, and thus are incompatible with interactive simulation environments based on engines like MuJoCo [30] or Isaac Sim [22]. 2)

---

*Equal contribution
†Corresponding author

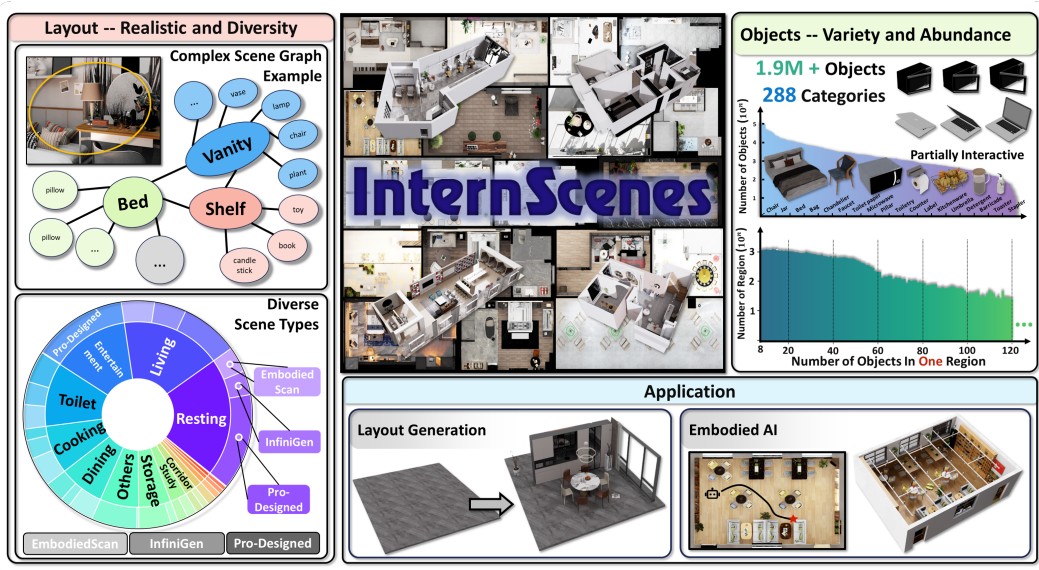

Figure 1: InternScenes is a large-scale, simulatable indoor scene dataset with diverse layouts and various 3D objects. It supports various tasks, such as scene layout generation and vision navigation.

Designer-created scenes [12, 40] feature a large number of simulatable 3D object assets. Nevertheless, these datasets deliberately omit small items [32], such as those on tables or cabinets, resulting in overly sanitized scenes that contradict realistic layouts. Furthermore, severe object collisions [35] are prevalent in these datasets, significantly hindering their integration into simulation environments. 3) Procedurally generated scenes [24], in theory, can offer an unlimited number of scenarios and avoid object collisions through delicately crafted rules. On the downside, these scenes are resource-intensive and time-consuming to generate, and often suffer from a lack of diversity. Ultimately, none of these datasets has adequately considered the inclusion of interactive objects.

In this paper, we introduce **InternScenes**, a large-scale, simulatable indoor scene dataset characterized by its diversity and realistic layouts. To ensure diversity, we integrate three distinct types of scene data: real-world scanned scenes from EmbodiedScan [32], procedurally generated scenes from Infinigen indoors [24], and designer-created synthetic scenes, correspondingly producing 3 subsets: *InternScenes-Real2Sim*, *InternScenes-Gen*, *InternScenes-Synthetic*. These diverse data sources have respective advantages: EmbodiedScan comprises small-scale single regions with realistic layouts, while Infinigen indoors provides various scenes with meticulously arranged and zero-collision object placement via subtle rules. In addition, the considerable designer-created synthetic scenes further offer extensive diversity and broader spatial coverage. To handle their different data formats and annotations, we customize corresponding data pipelines to make them simulation-ready. For EmbodiedScan, we create simulatable replicas for real-world scenes by replacing scanned objects with suitable object assets retrieved from Objaverse [11]. It is noteworthy that EmbodiedScan contains extensive annotations of small objects, allowing us to preserve realistic layouts after the real-to-sim transformation. To maintain realistic and complex indoor layouts, we select designer-created scenes with a large number of objects, particularly including numerous small items, and advocate object-number-aggressive rules while obtaining scenes via Infinigen indoors [24].

Consequently, as shown in Figure 1 our dataset consists of approximately $40,000$ diverse indoor scenes, including 48k regions from 15 common types in daily life, *e.g.*, living region, resting region, dining region, *etc.*, and features 1.96M objects and 800k CAD models covering a comprehensive taxonomy of 288 object classes within indoor scenes. Furthermore, we substitute roughly $20\%$ 3D assets inside with interactive objects from PartNet-Mobility and subsequently put all the scenes into the physical simulator to prevent object collisions, yielding a large-scale dataset of simulatable scenes with complex and realistic layouts. For example, each region of InternScenes has the highest-ever average number of 41.5 objects. Table 1 provides an overall comparison between InternScenes and existing 3D indoor scene datasets.

Table 1: Comparison with other 3D indoor datasets, where "-" represents "not available" or "not reported", "∞" indicates unlimited generation capability but requires significant time and computational resources. "Avg. Objects" indicates the average objects per region.

| Dataset | Layout Type | #Scenes | #Regions/ Reg.Types | #Objects/ Obj.Types | #Avg. Objects | #CAD Models | Physical Optimization |
|---|---|---|---|---|---|---|---|
| MP3D[5] | Real | 90 | -/- | 50K/40 | - | - | ✗ |
| EmbodiedScan[32] | Real | 9588 | 16K/12 | 230K/288 | 14.4 | - | ✗ |
| Structured3d[40] | Designed | 3.5K | 21K/- | 444K/40 | 21.1 | - | ✗ |
| Hypersim[26] | Designed | 461 | -/- | 58K/40 | - | - | ✗ |
| 3D-front[12] | Designed | 6.8K | 19K/8 | 140K/49 | 6.9 | 13K | ✗ |
| Behavior-1K[18] | Designed | 50 | 373/8 | -/1949 | - | 9K | ✓ |
| SceneVerse[16] | Real+Designed | 68K | -/- | 1.5M/- | - | - | ✗ |
| ASE[2] | Generation | 100K | -/- | -/29 | - | 8K | ✗ |
| Infinigen[24] | Generation | ∞ | -/- | ∞/89 | - | ∞ | ✗ |
| InternScenes | Real+Designed +Rule-based | 40K | **48K/15** | **1.96M/288** | **41.5** | **800K** | ✓ |

To fully harness the potential of **InternScenes**, we preliminarily use it for two benchmark applications: scene layout generation and point-goal visual navigation. First, we build two versions of InternScenes for scene layout generation: a full version with all objects included and a simplified version with all the small objects removed. Although trained with this large-scale dataset, current state-of-the-art methods still have unsatisfactory performance on the full version. It indicates the challenging nature of such complex scene generation, appealing to new model paradigms in the future. Furthermore, thanks to the simulation-ready property of InternScenes, we build the point-goal visual navigation benchmark to apply it for embodied AI. The complex and cluttered environments also pose great challenges for previous navigation policies. More importantly, we further generate more episodes from the diverse scene assets, and the experiments demonstrate the data's efficacy in boosting the generalization of our policies. We will open-source InternScenes with its corresponding data pipelines and benchmarks to the community, and hope they can pave the way from simulation to real-world applications for both AIGC and embodied AI algorithms.

## 2  Related Work

**Real-world Scans of Indoor 3D Scenes.** To directly obtain information from the 3D world for perception, researchers have employed various sensors to scan real environments, capturing RGB-D images that are subsequently reconstructed and annotated. Datasets [8, 31, 5, 37, 28, 25] such as ScanNet, MP3D, and 3RScan are utilized to enhance models' 3D perception capabilities. While direct scanning retains substantial real-world information, it is constrained by the complexity of the data acquisition process and limitations of the collection equipment. These factors frequently introduce unavoidable noise, thereby posing significant challenges to scene perception.

In recent years, researchers have made significant progress in enhancing reconstruction quality and annotation precision. For example, ScanNet++ [37] utilizes higher precision equipment compared to ScanNet [8] to achieve improved reconstruction and semantic annotation. EmbodiedScan [32] has enriched datasets from ScanNet, MP3D, and 3RScan with extensive annotation information, including annotations for small objects within scenes. It expands the object categories to 288 and provides annotations with 9-DoF bounding box information. Despite employing higher-precision collection equipment and more detailed annotations, there still exists a considerable gap between these scenes in simulation environments and real-world scenarios, along with inevitable annotation errors. Constructing a large number of finely annotated real-to-sim scenes is labor-intensive and time-consuming. As a result, researchers are increasingly focusing on indoor simulation scenes.

**Simulated Indoor Scenes.** To efficiently and cost-effectively obtain large-scale, detailed indoor scene data, researchers increasingly rely on computer software to construct and process synthetic indoor scenes [19, 40, 26, 12, 18, 10, 2], enabling the acquisition of multimodal data from diverse viewpoints. However, due to copyright restrictions, researchers often cannot access the original 3D

assets directly and must rely on pre-rendered datasets. Although Hypersim [26] provides a processing pipeline that allows users to purchase the original assets and follow the whole pipeline for custom rendering, this approach is prohibitively expensive, with costs reaching approximately $57K.

As an alternative, rule-based generative models such as SceneScript [2] and Infinigen [24] can automatically generate an unlimited amount of 3D scene data via scripting. However, they are computationally intensive and time-consuming, and the resulting scenes often suffer from limited diversity. Another distinct approach is taken by 3D-FRONT [12], which has released 18K curated scene layouts and 13K CAD models, allowing researchers to reconstruct complete indoor scenes for novel tasks. However, since these layout designs are generated by learning from professional designers' inspirations, they often lack realism and diversity, resulting in scenes with fewer objects and missing many small items that are common in real environments. In contrast, InternScenes comprises approximately 40K diverse indoor scenes, encompassing 48K regions across 15 common daily-life categories. Each region contains an average of 41.5 objects, indicating a high density of objects within our scenes, including small items. Moreover, around 30% of the objects in each region are interactive.

**Real-to-Sim 3D Scene Generation.** To construct scene datasets with authentic spatial distributions suitable for physics-based simulation and embodied AI training, researchers typically employ a real-to-sim paradigm to assemble scene datasets. In this approach, real environments are scanned to acquire detailed layout information, which is subsequently transformed into synthetic scene assets. For instance, the OpenRooms [20] dataset builds upon ScanNet [8] indoor point cloud; it aligns ShapeNet [6] CAD models with the scanned furniture by the Scan2CAD [1] method. During this process, each object's bounding box is meticulously refined to enforce orthogonality with both the floor and wall planes and to remove any floating or intersecting artifacts, resulting in physically coherent scene layouts and object placements. Concurrent work MetaScenes [38] also builds delicate Real2Sim replica via replacing objects in ScanNet [8] Scenes, but its coverage is limited in a single data source. In contrast, certain methods forego point-cloud acquisition entirely, instead inferring spatial priors directly from a single image to synthesize 3D scenes. MIDI [15] conditions on a single image to generate multiple object assets in one pass. However, it frequently introduces visible artifacts and suffers from severe entanglement among objects of disparate scales, undermining realistic interactions. ACDC [9] method leverages vision-language models to extract scene distributions from a single image and reconstruct environments using BEHAVIOR-1K [18] assets. Despite its promise, ACDC struggles to accurately represent complex scenes populated with numerous small objects, limiting its fidelity in such scenarios.

## 3 Dataset

In this section, we detail our two-stage pipeline to build a diverse and realistic scene dataset. In the first stage, scenes from multiple sources are integrated and cleaned to extract layout information, while a diverse 3D asset library is curated to ensure accurate object-layout correspondence. In the second stage, objects are placed into scenes based on extracted layouts, followed by optimization and physics simulations to resolve issues such as collisions. Finally, we conduct a statistical analysis on our dataset, highlighting its quality and advantages.

### 3.1 Multi-Source Data Processing

**InternScenes-Real2Sim: Real-to-Sim Replica Creation on Real-world Scanned Scenes.** Currently, retrieving scenes from real to simulation environments still faces two core challenges. First, the layouts in real-world scenes exhibit high diversity and complexity, as residents often have personalized preferences for object arrangements within scenes. Second, real scenes commonly contain a large number of small objects, which are more heterogeneous in category, greater in quantity, and display significantly varied poses compared to large furniture items. To address these challenges, we propose an effective retrieval pipeline, illustrated in Figure 2. The detailed annotations of regions and numerous small objects in EmbodiedScan precisely meet our requirements, making it a valuable data source. To ensure sufficient object density within each region, we defined a set of rules to merge small, semantically similar regions, guaranteeing that each resulting region contains at least 8 objects.

To further cover all object categories present in EmbodiedScan and to enable interactive capabilities in the retrieved scenes, we perform label mapping and canonical pose correction on raw assets from

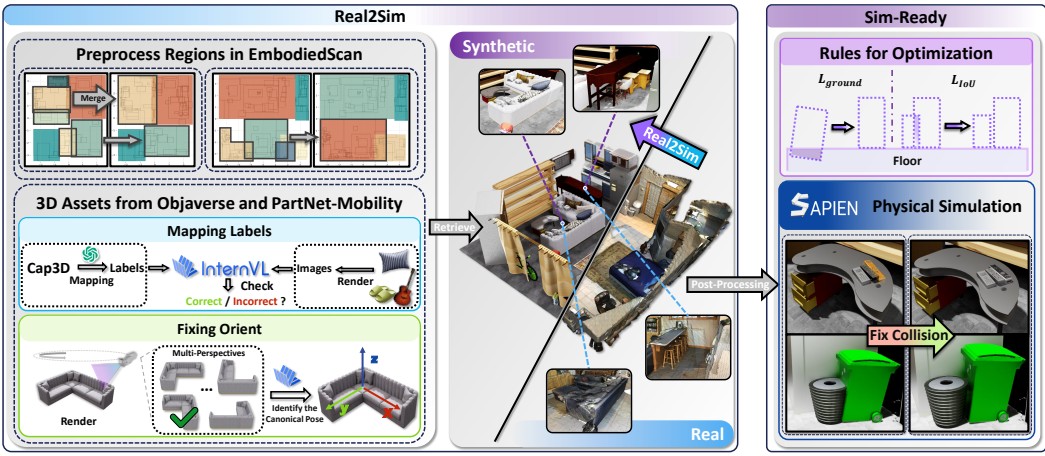

Figure 2: Pipeline for retrieving synthetic scenes from real scan scenes

Objaverse [11] and PartNet-Mobility [35]. For label mapping in Objaverse, we utilize GPT-4o to map descriptions from Cap3D [21] into 288 predefined categories. The mapping results, along with their corresponding rendered images, are then fed into the InternVL [7] model for verification and filtering to eliminate incorrect mappings. In contrast, the label system in PartNet-Mobility is relatively limited, so we conducted manual label matching.

For objects with orientation constraints, we further perform canonical pose correction. Specifically, we render such objects from multiple viewpoints at an oblique top-down angle and input these renderings into InternVL. The model identifies and outputs the index of the image that best represents the front-facing view, based on which we align the main direction of each object to the positive x-axis with the Euler angles annotated in EmbodiedScan. To more effectively align object arrangements with their corresponding assets in real-world scenarios, we further propose a candidate object selection mechanism coupled with a fuzzy label replacement strategy. A comprehensive description of the underlying rules is provided in the supplementary material.

**InternScenes-Gen: Procedurally Generated Scenes constrained by Rules.** We also include scene layouts generated by Infinigen Indoors, which is a procedural generator for creating photorealistic indoor scenes. It employs a constraint-based arrangement system. It defines scene composition constraints for several region types through a domain-specific language. These constraints cover various aspects such as symmetry, spatial relations, quantity, physics, and accessibility. The system then employs a solver to generate scene compositions that maximally satisfy these constraints. The scenes generated by Infinigen Indoors are photorealistic and semantically plausible. It can generate complex indoor scenes with object arrangements that adhere to physical and functional constraints, and it is capable of generating detailed indoor settings, such as dining tables with various objects and items inside cabinets. We implement relevant algorithms to extract and save scene layouts from the scenes generated by Infinigen Indoors.

**InternScenes-Synthetic: Annotation for Designer-Created Scenes.** The organization of synthetic scenes should ideally follow a logical sequence, progressing from general to specific elements, thereby establishing a hierarchical structure of *Scene-Regions-Instances-Parts*. This clear hierarchical division enables efficient extraction and understanding of information related to the scenes and their constituent objects. However, in practice, the data structures created by designers frequently display disorganized arrangements and insufficient annotation, which present notable challenges for data collection.

Specifically, at the *Regions* level, a single house typically contains multiple functional regions, yet these regions are not clearly delineated. For example, in an apartment suite, the living region and cooking region might coexist in the same scene, but designers often fail to distinctly define their boundaries, rendering it impossible to implement automatic segmentation using standard algorithms. At the *Instances* level, there are both furniture sets that are physically combined in the scene and parts that theoretically should be combined but are not. For instance, a sofa and the pillows or magazines placed on it are defined by the designer as a single instance, while the table legs and tabletop are

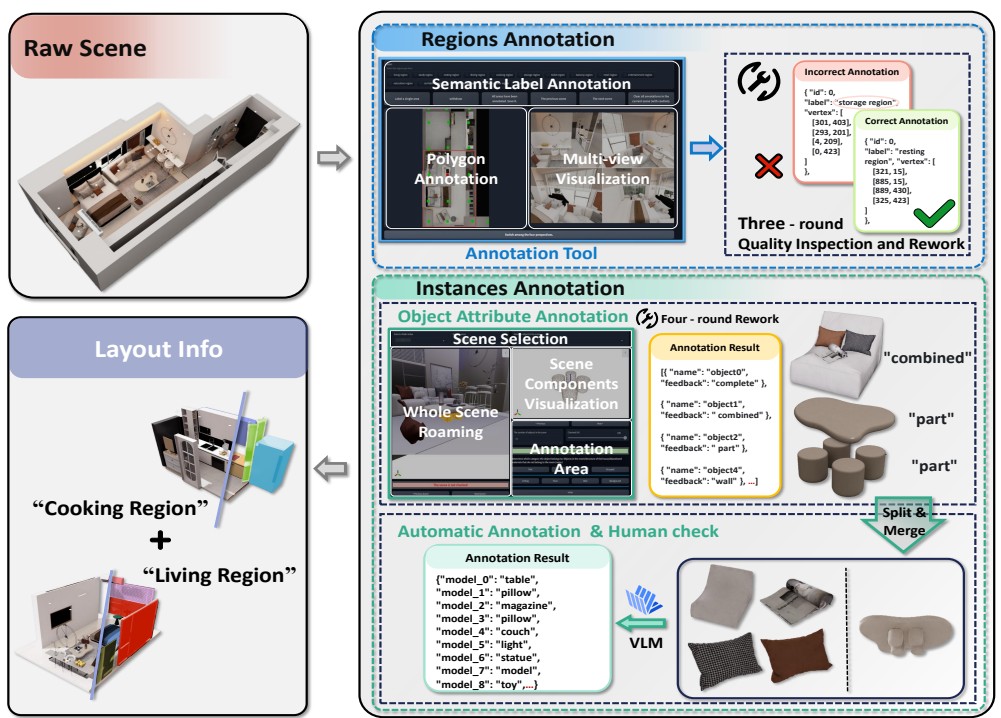

Figure 3: Pipeline for annotating and processing raw scenes to extract precise layout information.

split into separate instances. This approach to organization not only leads to significant ambiguity in semantic instance judgment but also results in potentially inaccurate bounding box dimensions. To address the aforementioned issues, we refined the definition of region types within the scenes and performed splitting or merging operations on instances to capture the finest layout distribution within each region. The overall processing pipeline is illustrated in Figure 3.

*Region Annotation.* Given the lack of a universal region segmentation algorithm, we adopted a manual annotation approach to define region types. To facilitate this process, we developed a dedicated region annotation tool consisting of three core modules. The Multi-View Visualization module displays multi-view renderings of sampled points within the scene. The Polygon Annotation module presents the bird's-eye-view map of the entire scene and allows annotators to delineate regions using polygonal drawings. The Semantic Label Annotation module enables annotators to assign semantic categories by selecting from a set of predefined semantic label options. After three rounds of annotation, review, and correction, we obtained the coordinate information and attribute labels for all regions in the scene.

*Instance Annotation.* To address the hierarchical disorder among objects in the original scenes, we relied on human judgment to determine whether objects needed splitting or merging. For this purpose, we built an instance annotation tool that allows annotators to freely navigate the 3D scene and locate target objects for evaluation and labeling. Based on these annotations, we wrote scripts to automatically perform object splitting and merging. Subsequently, we rendered the processed objects from six different viewpoints and fed these images into the InternVL to generate semantic labels automatically, which are then verified by human annotators. Based on the region and instance annotation results, we further extract the object coordinates, bounding box dimensions, and rotation Euler angles within each region, thereby forming the necessary layout information.

## 3.2 Physics-Aware Scene Composition

To prevent collision and clipping issues between objects in the scene, we perform physical simulation optimization on the scenes obtained in the previous step. Specifically, we first conduct fine-tuning of the bounding boxes of the objects and then place them into a simulator to perform final computational adjustments to achieve the final scene layout.

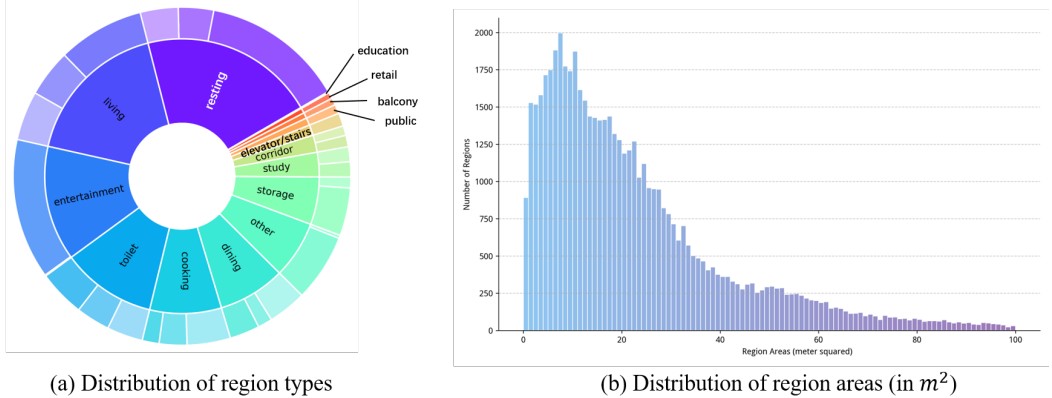

(a) Distribution of region types

(b) Distribution of region areas (in $m^2$)

Figure 4: **Region statistics.** Our dataset includes 15 common scene categories, such as the resting room and the living room. We also show the distribution of region areas.

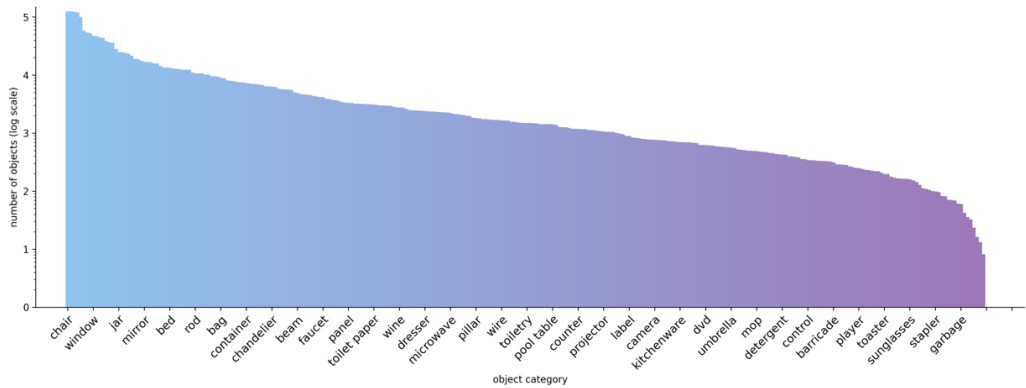

Figure 5: Distribution of objects across 288 categories. We list the 30 categories with the highest occurrence frequency.

**Bounding Box Optimization and Fine-Tuning.** Given the distribution characteristics of objects within a region, we establish different rules for larger furniture objects and smaller object assets. For smaller assets, we first bind their positions to nearby larger objects to ensure that the relative positions of large furniture and smaller object assets remain stable during the fine-tuning process. For larger furniture, we implement a loss function composed of three parts: $L_i = L_{\text{IoU}} + L_{\text{ground}} + L_{\text{reg}}$. The IoU Loss is used to optimize overlapping and clipping among large furniture items. The Ground Loss addresses noise introduced during scanning, which can lead to misalignment of objects with the floor. The Regularization Term ensures that these objects do not deviate significantly from their original positions.

**Simulator Processing.** To further enhance the physical plausibility of smaller objects in the region and avoid common issues such as object collision and floating, we import the optimized furniture layout into the SAPIEN [35] engine for detailed physics simulation, after completing the bounding-box-based optimization for large furniture items. Specific implementation details regarding bounding box optimization and physics simulation can be found in the supplementary material.

### 3.3 Dataset Statistics

**Region and Object Statistics.** Our dataset comprises three subsets, totaling 39870 scenes and 48381 regions across 15 categories. Specifically, the InternScenes-Real2Sim subset contains 9833 regions, InternScenes-Gen contains 11454 regions, and InternScenes-Synthetic contains 27094 regions. In total, 1.96M objects from 288 categories are placed across all regions, sampled from our asset library of 80M CAD models. These objects are sampled from our asset library containing 80 million CAD

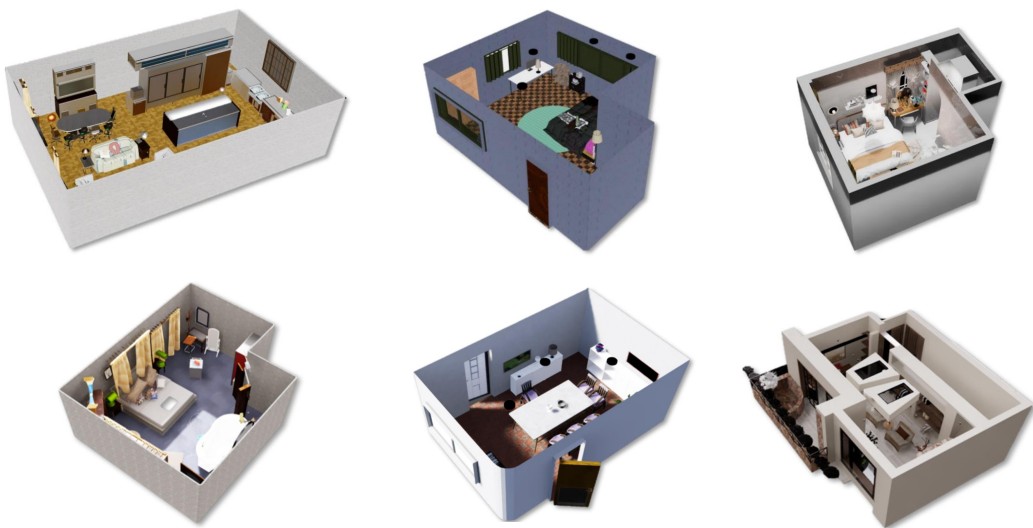

Figure 6: Scene Examples in *OmniScenes-Real2Sim*(left), *Gen*(middle), *Synthetic*(right)

models. On average, each region contains 41.5 objects. Figure 4 (a) shows the distribution of 15 region types. Figure 4 (b) shows the distribution of region area (in m$^2$). Figure 5 shows the distribution of objects across 288 categories. The five most frequent object categories are *chair*, *toy*, *book*, *light*, and *bottle*. In Figure 6, we provide some scene examples of InternScenes for visualization.

**Object Relation Statistics.** Following [39], we quantified the number of containment and support relationships between objects in the InternScenes dataset to characterize its structural complexity. Specifically, we selected a subset of object categories from InternScenes—categories that are not only commonly present in indoor scenes but also likely to serve as containers or supporting structures. For all objects falling into these selected categories, we calculated the total number of containment and support relationships, which were determined by the presence of smaller objects either inside (for containment) or on top of (for support) them. On average, each of these objects contains or supports 3.45 other objects. When excluding the subset of objects that exhibit no containment or support relationships, this average rises to 5.57 objects per supporting or containing object.

## 4 Experiment

This section presents two preliminary benchmarks built upon InternScenes to show its application in 3D AIGC and embodied AI. Specifically, given the complex and diverse layouts provided in InternScenes, we first introduce an interior scene generation benchmark and show the new challenges posed by the large number of small objects involved (Section 4.1). Subsequently, since InternScenes is simulation-ready, we use it to benchmark point-goal navigation methods and discuss the new challenges caused by more realistic, cluttered environments in Section 4.2.

### 4.1 Interior Scene Generation

The first important property of InternScenes is its complex and realistic layout, which bridges the gap in the field of scene generation. Therefore, we first build an interior scene generation benchmark to validate the efficacy of our dataset and study the emerging challenges.

**Dataset Construction.** We selected three common region types from the InternScenes dataset, Resting, Living, and Dining regions, for our interior scene generation experiments. To decouple the effect of the large number of small objects in InternScenes, we construct two versions of datasets for different difficulty levels: 1) Full Version that includes all objects, and 2) Simplified Version that removes all small objects. Then we use all the scenes for training the generative baseline models.

Table 2: Quantitative evaluation results of ATISS[23], DiffuScene[29], and Physcene[36] trained separately on the full and simplified versions of the Internscenes dataset. For SCA, the score closer to $50\%$ is better. Lower FID and CKL demonstrate better generation performance.

| Dataset | Method | Resting Region | | | Living Region | | | Dining Region | | |
|---------|--------|---------|------|---------|---------|------|---------|---------|------|---------|
| | | FID($\downarrow$) | SCA% | CKL($\downarrow$) | FID($\downarrow$) | SCA% | CKL($\downarrow$) | FID($\downarrow$) | SCA% | CKL($\downarrow$) |
| Full Version Dataset | ATISS | 101.85 | 95.65 | 0.178 | 104.48 | 96.95 | **0.091** | 133.20 | 99.44 | 0.151 |
| | DiffuScene | 96.56 | 95.40 | 0.232 | 107.49 | 96.66 | 0.149 | **122.95** | **97.54** | 0.235 |
| | Physcene | **88.02** | **94.97** | **0.175** | **66.59** | **96.45** | 0.123 | 130.39 | 98.91 | **0.081** |
| Simplified Version Dataset | ATISS | 23.20 | 59.80 | 0.133 | 30.49 | 70.95 | 0.056 | 30.89 | 64.72 | 0.063 |
| | DiffuScene | **22.88** | **57.70** | **0.117** | **23.54** | **64.30** | **0.057** | 28.70 | **59.99** | 0.095 |
| | Physcene | 23.78 | 68.45 | 0.142 | 24.75 | 64.40 | 0.058 | **26.76** | 66.82 | **0.047** |

**Experimental Setup.** We employed the unconditional generation mode for all three baseline models. To ensure a fair comparison, we retrained a Variational Autoencoder (VAE) for point cloud compression using InternScenes assets and mapped the original object categories to our defined 288-category taxonomy. Performance was evaluated on $1,000$ generated scenes using four common metrics in indoor scene generation: Fréchet Inception Distance (FID) [14], Kernel Inception Distance (KID × 0.001) [3], Scene Classification Accuracy (SCA), and Category KL Divergence (CKL × 0.01). For FID, KID, and SCA metrics, we rendered a 256×256 resolution orthographic top-down view for each real and generated scene. We benchmark three representative baseline methods for analysis, namely ATISS [23], DiffuScene [29], and PhyScene [36].

**Results and Analysis.** The quantitative results of our experiments are presented in Table 2. A comparison of the different baselines, when trained on identical datasets, reveals that DiffuScene and PhyScene generally exhibit superior performance across most metrics. This observation aligns with the performance distribution of these baselines on the 3D-FRONT [12] dataset, which indirectly substantiates the plausible realism of the InternScenes dataset.

However, when employing the same methodology and experimental setup but utilizing distinct training data, all three baselines demonstrate a decline in performance on the complete version of InternScenes. Our findings indicate that while the three baselines perform commendably in generating indoor scenes composed of large furniture items, they encounter difficulties in capturing the extensive array of small objects characterized by complex distributions within the comprehensive dataset. This highlights that the placement of small items is not a trivial task and constitutes a challenging problem as explored by works on micro-scene generation [38, 13].

Furthermore, on the simplified version of the InternScenes data, the results obtained by DiffuScene and PhyScene are largely comparable across most metrics. Conversely, in the context of complex scenes within the complete dataset, PhyScene exhibits a pronounced advantage over DiffuScene. This suggests that the physics-based guidance mechanism integrated into the PhyScene method may potentially boost the efficacy of diffusion-based scene generation algorithms in producing physically plausible and complex scenes.

## 4.2 Navigation

Next, we choose point-goal navigation as the benchmark application of InternScenes for embodied AI. Previous scene datasets for point-goal navigation either have simple layout complexity or limited diversity. In contrast, InternScenes provides diverse simulation-ready environments that can generate considerable episodes therein. More importantly, it offers a challenging testbed for testing point-goal navigation algorithms in diverse, realistic, cluttered scenes.

**Experiment Setup.** To evaluate the efficacy of our scene datasets for downstream Embodied AI tasks, we build a physically and visually realistic point-goal navigation benchmark based on IsaacSim and our scene assets, which distinguishes from prior physical-agnostic navigation benchmarks, such as Habitat-Sim [27] and AI2Thor [17]. For a more comprehensive investigation of the sim-to-real gap in navigation approaches, we manually select 20 scenes from InternScenes-Real2Sim and 10 from InternScenes-Gen, with selection criteria based on layout complexity and asset quality. The wheeled robot ClearPath Dingo is adopted as the navigation agent. Two metrics are employed in

Table 3: The PointGoal navigation benchmark results across different baseline methods.

| Method | InternScenes-Real2Sim | | | InternScenes-Gen | | |
|---|---|---|---|---|---|---|
| | Success(↑) | SPL(↑) | Distance(-) | Success(↑) | SPL(↑) | Distance(-) |
| DD-PPO [34] | 23.6 | 23.1 | 5.41 | 45.0 | 44.2 | 4.94 |
| NavDP [4] | 48.3 | 45.3 | - | 61.9 | 61.8 | - |
| NavDP-FT [4] | 51.0 | 49.4 | - | 63.6 | 61.7 | - |

the benchmark: Success Rate and Shortest Path Length (SPL). Success Rate assesses whether the agent can find a valid path to reach the goal, while SPL measures the efficiency of the executed path relative to the oracle shortest path. Each scene is evaluated across 20 episodes, and we report the average distance between all starting points and target points to quantify task difficulty.

**Baseline.** Three representative baseline methods are considered in the evaluation. The first is an RL-based method, DD-PPO [34], which is massively trained in Habitat-Sim [27]. As DD-PPO trains the policy with respect to a discrete action space, we deploy it in the continuous action space by multiplying the discrete predicted coordinates with a coefficient into linear and angular speed. The second is a pretrained diffusion-based imitation learning method, NavDP [4]. The third is a fine-tuned version of NavDP. To fine-tune the NavDP, we follow their data generation pipeline with our Internscenes assets and compose a new navigation dataset with 118,784 trajectories.

**Results and Analysis.** Navigation performance metrics are reported in Table 3. DD-PPO [34] achieves a low success rate across all scenes, indicating that RL-based policies have limited generalization capabilities when confronted with continuous action spaces and domain gaps during motion execution. While NavDP [4] can select optimal trajectories using its pretrained critic function and complete navigation tasks in many scenarios, the cluttered layouts in InternScenes pose unique challenges, leading to a success rate of approximately 50%. By fine-tuning NavDP with additional navigation trajectories from InternScenes, we observe a slight improvement in its overall performance. This result demonstrates that the diversity of our InternScenes dataset can facilitate model training; however, how to scale model capacity to leverage increasingly large datasets remains an open problem.

**Discussion and Conclusion.** Based on the navigation performance metrics, we observe a significant performance decline in our evaluation framework. By analyzing failure cases in depth, we identify three key challenges in our benchmark and propose a potential direction for future research on navigation methods. First, the realistic scene assets in our benchmark tend to feature cluttered room layouts, which demand more precise path-planning capabilities and collision recovery mechanisms. The lack of collision recovery capabilities in the baseline methods is a key factor contributing to their performance degradation in cluttered environments. Second, our scene assets frequently include narrow pathways, where traversability depends entirely on the robot's embodiment information. However, most learning-based navigation methods rely solely on exteroceptive observations, which constrains their navigation performance in such scenarios. Third, real-world objects often have small connected components (e.g., office chair legs) that are classified as obstacles. While these tiny obstacles may be captured in visual observations only in limited frames, they are critical for safe path planning—posing a major challenge to the spatial perception capabilities of navigation approaches. These three characteristics make our navigation benchmark an ideal platform for evaluating the sim-to-real gap of navigation methods.

# 5 Limitations and Conclusion

In this work, we introduce **InternScenes**, a large-scale, simulatable indoor scene dataset with diverse and realistic layouts, constructed by integrating real-world scans, procedural generation, and synthetic design. Featuring 40,000 scenes and over 1.96 million objects from 288 classes, InternScenes enables new benchmarks in layout generation and visual navigation, posing significant challenges to current methods. We open-source the dataset and tools to support future research in embodied AI and AIGC. Although this paper presents a pipeline for processing multi-source scene data, the current approach remains reliant on manual annotation and can be further improved regarding scene diversity. Future work will aim to reduce human involvement and further improve the quality of the 3D assets library.

## 6 Acknowledgement

This work is funded in part by the National Key R&D Program of China (2022ZD0160201), Shanghai Artificial Intelligence Laboratory, and HKU Startup Fund.

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

# A  Pipeline Details

This section supplements several details in the two-stage pipeline, mainly including the retrieval details of *InternScenes-Real2Sim* and the annotation details of *InternScenes-Synthetic* in the first stage and details of *Physics-Aware Scene Composition* in the second stage.

## A.1  Retrieval Details of *InternScenes-Real2Sim*

**Object Category Replacement and Candidate Asset Selection Strategy.** In the retrieval process, our goal is to find and match the most suitable 3D object instance for each bounding box in the EmbodiedScan [32] dataset from a pre-curated 3D asset library and place it in the corresponding location within the scene. For object categories with clear semantic definitions, their candidate assets are directly composed of all available instances under that category in the asset library.

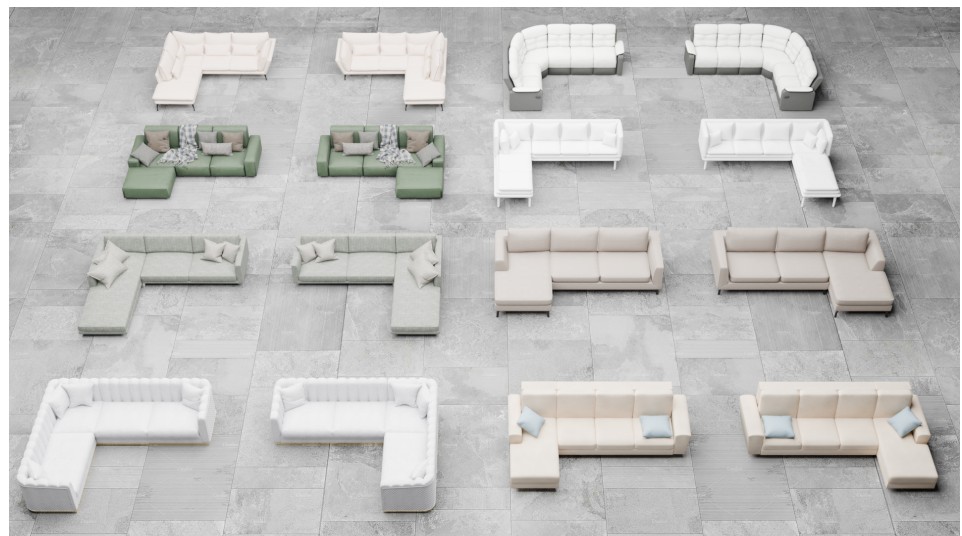

Figure 7: Examples of symmetrical L-shaped couches.

However, some categories in EmbodiedScan are defined too broadly or ambiguously, potentially covering multiple specific subcategories. For example, the category "object" might refer to items such as books, plants, or lamps placed on a desk, or it could represent small objects like shoes located on the floor. To address such semantically ambiguous categories, we introduce a context-based rule-driven label replacement mechanism. Specifically, by analyzing the spatial position of an object labeled as "object" within the scene and the semantic information of its neighboring objects, we infer a more specific alternative category.

For instance, if an "object" is located on a desk, its semantic category can be further refined into one of several predefined categories, such as "book," "plant," or "lamp". In this case, the set of candidate assets for that object will consist of all 3D models under these refined categories in the asset library. The complete mapping rules from ambiguous to specific categories are detailed in Table 4.

Building upon the category replacement for "object", we further consider the special shape requirements of objects within scenes. Take L-shaped couches as an example—these may exhibit two distinct spatial configurations: left-L and right-L (mirror-L). Based on the spatial distribution of bounding boxes in the scene, we classify couches into three types: left-L, right-L, and standard (non-L). Due to limited diversity in specialized shapes within the asset library, we manually group existing couch models into these three categories and apply mirror symmetry transformations to the left-L and right-L types, allowing them to complement each other in different scenarios, which enhances both the adaptability and variety of candidate couches in terms of shape. The L-shaped couches are illustrated in Figure 7.

**Select from candidate assets.** For a given object in the scene, we select the asset that best matches the annotated bounding box dimensions provided by EmbodiedScan [32] from all its candidate assets. By introducing bounding box similarity as an evaluation metric, we can effectively reduce morphological distortions caused by scale stretching.

Table 4: Substituted Object Categories by Position

| Object Position | Substituted Categories |
|---|---|
| on floor | "bin", "bag", "backpack", "basket", "shoe", "ball" |
| on bed / couch | "toy", "pillow", "bag", "book", "backpack", "hat" |
| on table / desk | "book", "plant", "lamp", "bottle", "socket", "cup", "vase", "bowl", "plate", "fruit", "teapot" |
| in washroom | "cup", "box", "bottle", "towel", "case", "soap", "soap dish", "soap dispenser" |
| in kitchen / on stove | "bowl", "cup", "knife", "plate", "can", "fruit", "food" |
| in / on cabinet | "box", "toy", "book", "hat", "bag", "cup", "shoe" |
| attached to wall | "picture", "socket" |

Due to the diverse origins of the assets, their scales are not uniformly aligned. Therefore, before computing the bounding box similarity, we first normalize the bounding box dimensions. Let the target bounding box size vector be $\mathbf{t} \in \mathbb{R}^3$, and the $i$-th candidate bounding box size vector be $\mathbf{c}_i \in \mathbb{R}^3$. Then, the bbox similarity is defined as:

$$\text{sim}(\mathbf{c}_i, \mathbf{t}) = \frac{\sum_{j=1}^{3} c_{i,j}\, t_j}{\sqrt{\sum_{j=1}^{3} c_{i,j}^2} \sqrt{\sum_{j=1}^{3} t_j^2}}$$

After the asset is selected, we transform the chosen 3D model according to the size, translation, and rotation information of the object's bounding box in the original scene, so as to accurately place it into the corresponding position within the scene.

## A.2   Annotation Details of *InternScenes-Synthetic*

**Region Annotation.** In our region annotation tool, achieving an effective perception of the overall scene environment and precise region annotation requires the use of the BEV map of the scene along with the corresponding sampling point information. To facilitate this, we employ IsaacSim [22] to render images from multiple perspectives within the scene, which supports the subsequent annotation processes.

To generate the BEV map, the process begins by converting the entire scene into point clouds and performing downsampling to extract a histogram of the z-axis height distribution. Next, the z-axis coordinates corresponding to the peaks in this histogram are identified. These coordinates, combined with the DBSCAN clustering method, help estimate the height range for the floor and ceiling. Finally, a *Rect Light* is positioned 1.5 meters above the floor, and an orthographic camera is placed 1.8 meters above the floor to capture the entire scene, resulting in a clearly structured BEV map.

To generate multi-view rendered images of specific sampling points, we start by downsampling the floor point cloud to determine the sampling locations. At each sampling location, a perspective camera is positioned 1.8 meters above the floor. The camera captures images by rotating around the point in 45-degree increments, resulting in a total of 8 different perspective rendering images. This comprehensive method ensures that all spatial information surrounding the sampling point is thoroughly captured. Figure 8 shows the BEV maps of some scenes along with the rendered images of their corresponding sampling points.

**Instance Annotation for Splitting and Merging.** Given the difficulty of accessing the original data format, we convert the entire scene into a mesh and transform all its constituent elements into point clouds with color information. This transformation facilitates easier access to the data.

During the annotation process, when a user selects a specific element, it is highlighted within the scene, and the camera view automatically adjusts to focus on that element. This adjustment helps users better understand the element's exact location within the scene, enabling more precise annotation operations.

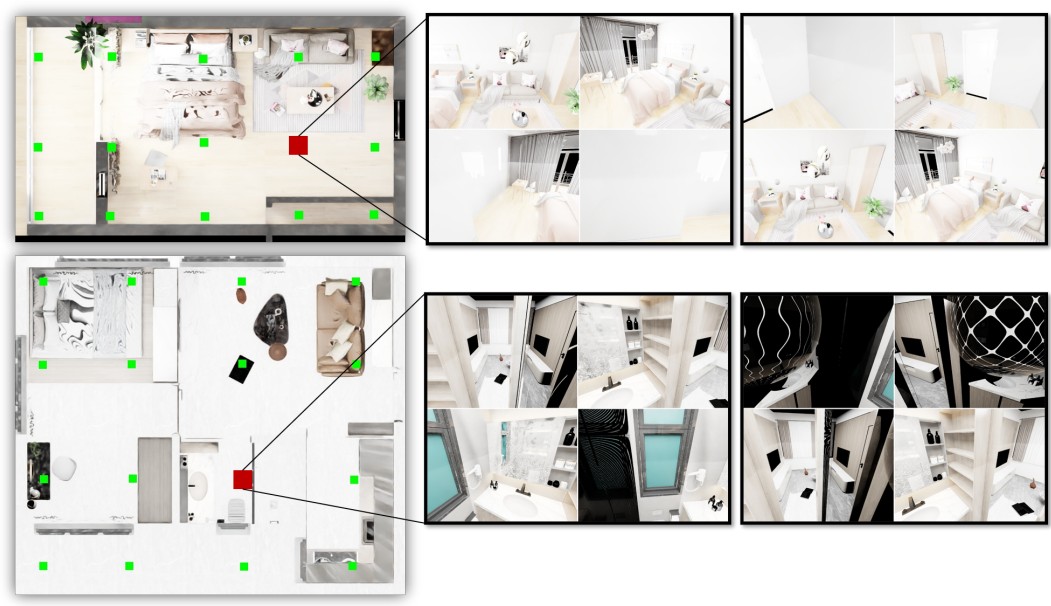

Figure 8: Examples of BEV maps and rendered images of their corresponding sampling points

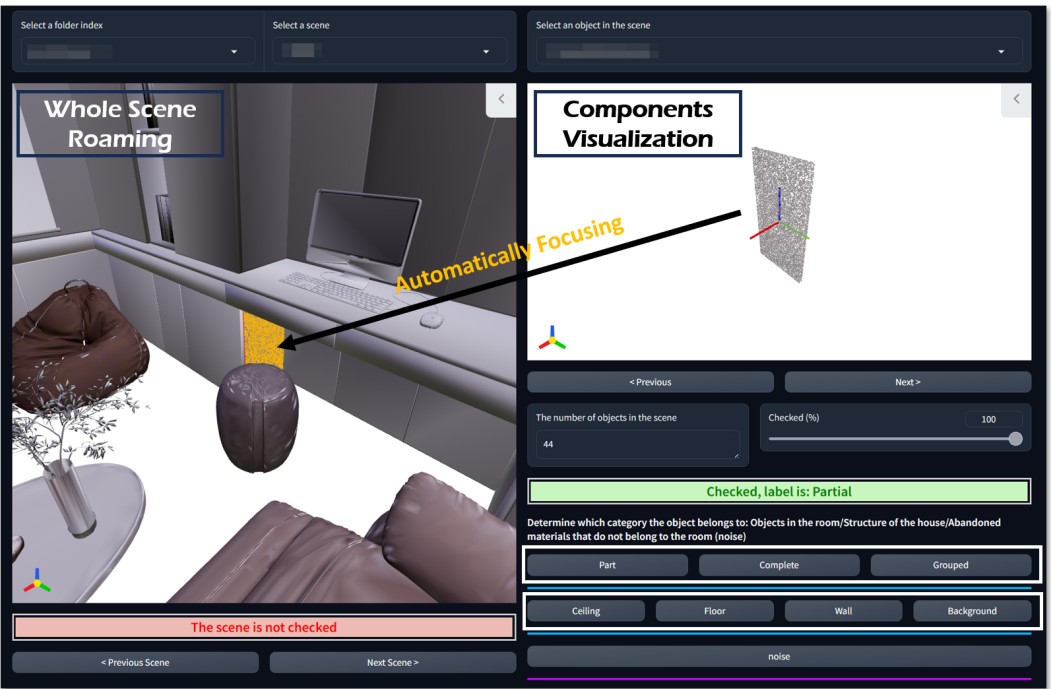

Figure 9: Instance annotation interface UI

For label selection, users can primarily choose from two major categories: object types and room structure types. Object type annotations are further divided into three subcategories: individual complete objects, which are standalone entities with clear semantic definitions; assemblies, which are sets or collections composed of multiple objects with different semantics; and partial objects, which represent components of a complete object. Room structure types are categorized into floor, ceiling, walls, and background, providing a more accurate description of the spatial composition within the scene. Figure 9 provides a detailed illustration of the user interface design for the annotation tool. Based on the annotation results, we perform automated splitting or merging of objects within the scene.

**Instance Annotation for Semantic Labels.** We extract the processed instance assets from the scene and utilize IsaacSim to render them from multiple viewpoints. Specifically, for both the 45-degree upper diagonal and 45-degree lower diagonal perspectives relative to the object, we perform three renderings at 120-degree intervals, resulting in a total of six views. These six rendered images are then collectively fed into the InternVL [7] model for automatic semantic annotation of the objects.

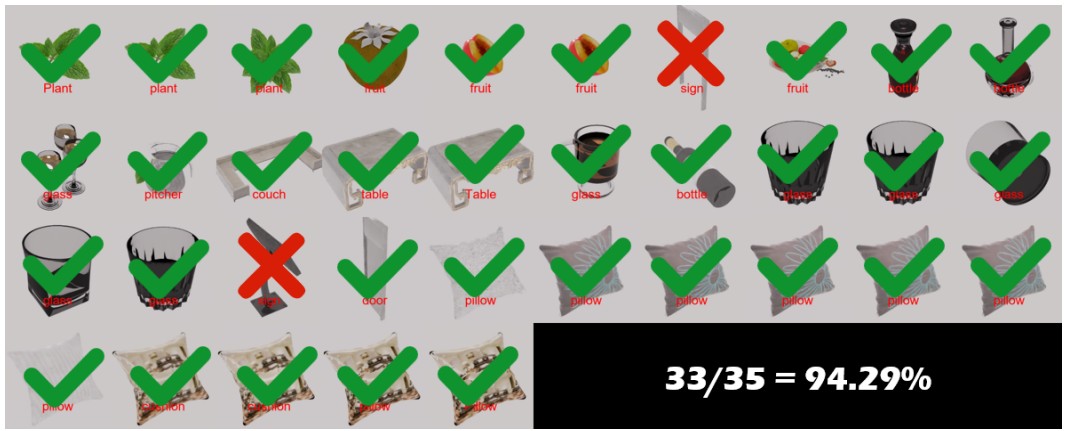

Figure 10: Inspection results of scene 4

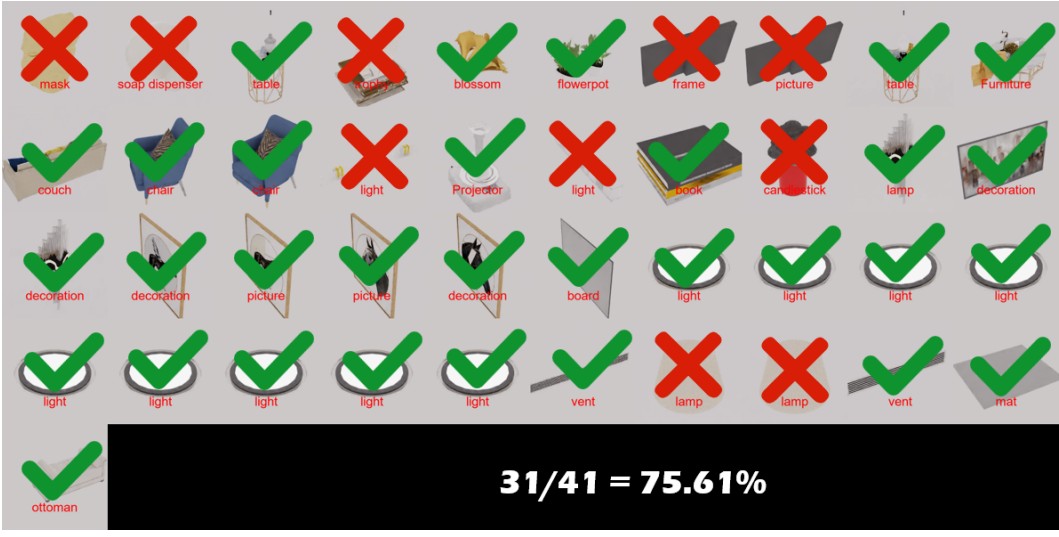

Figure 11: Inspection results of scene 9

Table 5: Automatic captioning accuracy for manual inspection

| ID | #Correct | #Incorrect | #All | Accuracy | ID | #Correct | #Incorrect | #All | Accuracy |
|----|----------|------------|------|----------|----|----------|------------|------|----------|
| 1 | 32 | 8 | 40 | 80.00% | 26 | 71 | 10 | 81 | 87.65% |
| 2 | 39 | 2 | 41 | 95.12% | 27 | 38 | 4 | 42 | 90.48% |
| 3 | 18 | 1 | 19 | 94.74% | 28 | 29 | 3 | 32 | 90.63% |
| 4 | 33 | 2 | 35 | 94.29% | 29 | 9 | 2 | 11 | 81.82% |
| 5 | 28 | 2 | 30 | 93.33% | 30 | 24 | 3 | 27 | 88.89% |
| 6 | 52 | 7 | 59 | 88.14% | 31 | 14 | 0 | 14 | 100.00% |
| 7 | 192 | 24 | 216 | 88.89% | 32 | 78 | 4 | 82 | 95.12% |
| 8 | 59 | 3 | 62 | 95.16% | 33 | 34 | 10 | 44 | 77.27% |
| 9 | 31 | 10 | 41 | 75.61% | 34 | 168 | 10 | 178 | 94.38% |
| 10 | 145 | 12 | 157 | 92.36% | 35 | 36 | 5 | 41 | 87.80% |
| 11 | 27 | 3 | 30 | 90.00% | 36 | 141 | 18 | 159 | 88.68% |
| 12 | 87 | 15 | 102 | 85.29% | 37 | 30 | 8 | 38 | 78.95% |
| 13 | 12 | 5 | 17 | 70.59% | 38 | 29 | 10 | 39 | 74.36% |
| 14 | 25 | 1 | 26 | 96.15% | 39 | 11 | 3 | 14 | 78.57% |
| 15 | 27 | 4 | 31 | 87.10% | 40 | 103 | 16 | 119 | 86.55% |
| 16 | 19 | 0 | 19 | 100.00% | 41 | 5 | 0 | 5 | 100.00% |
| 17 | 17 | 5 | 22 | 77.27% | 42 | 71 | 15 | 86 | 82.56% |
| 18 | 68 | 9 | 77 | 88.31% | 43 | 19 | 2 | 21 | 90.48% |
| 19 | 28 | 5 | 33 | 84.85% | 44 | 27 | 9 | 36 | 75.00% |
| 20 | 69 | 3 | 72 | 95.83% | 45 | 11 | 1 | 12 | 91.67% |
| 21 | 43 | 7 | 50 | 86.00% | 46 | 31 | 3 | 34 | 91.18% |
| 22 | 47 | 12 | 59 | 79.66% | 47 | 26 | 7 | 33 | 78.79% |
| 23 | 13 | 1 | 14 | 92.86% | 48 | 40 | 8 | 48 | 83.33% |
| 24 | 25 | 5 | 30 | 83.33% | 49 | 15 | 2 | 17 | 88.24% |
| 25 | 23 | 4 | 27 | 85.19% | 50 | 27 | 6 | 33 | 81.82% |

| | | | | |
|-----|------|-----|------|--------|
| All | 2237 | 313 | 2550 | 87.73% |

Finally, we conducted random inspections on a total of 2550 objects across 50 randomly selected scenes to evaluate the accuracy of the annotations. Figure 10 11 show the inspection results for some of the annotated objects, while Table 5 summarizes the distribution of label accuracy across these 50 scenes. The accuracy of automatic captioning can reach more than 85%

### A.3 Details of *Physics-Aware Scene Composition*

**Oriented Bounding Box Optimization and Fine-Tuning.** We optimize the oriented bounding box (OBB) position of large furniture, focusing on addressing issues such as furniture penetration or unreasonable interaction between furniture and the ground. To achieve this, we designed a loss function consisting of three terms:$\mathcal{L}_{IoU}$, $\mathcal{L}_{ground}$, and $\mathcal{L}_{reg}$, which are used to quantitatively evaluate the furniture layout. We represent the $N$ bounding boxes of the large furniture in the scene as a list $\{b_i\}_{i=1}^{N}$. The center translation of each bounding box $b_i$ is denoted by $t_i$, and we use $h_{\text{ground}}$ to denote the ground height. The overall loss function is as follows:

$$\mathcal{L} = \lambda_{\text{IoU}}\mathcal{L}_{IoU} + \lambda_{\text{ground}}\mathcal{L}_{ground} + \lambda_{\text{reg}}\mathcal{L}_{reg}.$$

Specifically, $\mathcal{L}_{\text{IoU}}$ prevents collisions by penalizing overlaps between objects. For any pair of large furniture items whose OBBs intersect, we compute the IoU of their Axis-Aligned Bounding Boxes (AABBs) as the loss value.

$$\mathcal{L}_{IoU} = \sum_{1 \leq j < k \leq N} \left[ \text{IoU}(b_j^{(t)}, b_k^{(t)}) \right]^2$$

The $\mathcal{L}_{\text{ground}}$ term ensures that the bottom surfaces of furniture items—such as sofas, chairs, and tables—stably align with the ground plane.

$$\mathcal{L}_{ground} = \sum_{j=1}^{N}(h_j^{(t)} - h_{\text{ground}})^2$$

Finally, $\mathcal{L}_{\text{reg}}$ restricts how much the furniture can deviate from its original annotated position during optimization, thereby preserving the spatial layout of the original scene while correcting physical inconsistencies. The overall optimization process is shown in algorithm 1.

$$\mathcal{L}_{reg} = \sum_{j=1}^{N}\left\|t_j^{(t)} - t_j^{(0)}\right\|_2^2$$

---

**Algorithm 1** OBB Optimization Algorithm

---

**Input:** Initial boxes $\{b_i^{(0)}\}_{i=1}^{N}$, Max iterations $T$, Ground height $h_{\text{ground}}$
**Output:** Final boxes $\{b_i^{(T)}\}_{i=1}^{N}$
1: initialize positions $\{t_i\}_{i=1}^{N} \leftarrow \{t_i^{(0)}\}_{i=1}^{N}$
2: **for** $t = 1$ to $T$ **do**
3:     $\mathcal{L}_i \leftarrow \text{ComputeLoss}\left(\{b_i^{(t)}\}_{i=1}^{N}, \{b_i^{(0)}\}_{i=1}^{N}, h_{\text{ground}}\right)$
4:     backpropagate and update $\{t_i\}_{i=1}^{N}$
5: **end for**
6: **return** $\{b_i^{(T)}\}_{i=0}^{N}$

---

**Simulator Processing.** After the bounding box optimization, the layout and physical plausibility of large furniture in the scene have been improved. However, small objects still exhibit artifacts such as floating or interpenetration. Moreover, due to the complex shapes of these small objects and the loose fit between the objects and their bounding boxes, further optimization using bounding box-based methods proves ineffective in resolving these issues. To address this, we employ physics simulation to refine the placement of small objects and eliminate such artifacts.

Prior to the physics simulation, we decompose each object in the asset library into convex collision primitives using the COACD [33] method. Notably, to enhance the realism of small object placements within scenes—particularly their ability to reside inside furniture with cavities (e.g., drawers or shelves)—we first perform a simple segmentation on cavity-containing furniture, breaking them into smaller components that expose the internal cavities. Each of these components is then individually processed with COACD decomposition. Finally, all resulting collision primitives are merged into a unified collision representation for the original object. This approach ensures that internal cavities are accurately captured in the convex collision geometry.

For the physics simulation, we utilize SAPIEN [35]. During the simulation, gravity and repulsive forces are enabled, allowing previously floating objects to settle naturally and interpenetrating objects to separate, ultimately yielding a physically plausible and realistic scene configuration.

# B  Experiments

This section supplements the details of two experiments mentioned in the main paper, including layout generation and navigation tasks.

## B.1  Interior Scene Generation

**Data and Implementation Details.** We conduct scene interior generation experiments using three commonly used regions from the InternScenes dataset: resting, living, and dining regions. Two versions of the dataset are constructed: a full version, which retains all objects present in the original InternScenes scenes, and a simplified version, which only preserves 45 large furniture object categories. The list of these categories is shown as follows:

```
# selected categories in simplified version dataset
["air conditioner", "bathtub", "beanbag", "bed", "bench",
"bicycle", "blinds", "cabinet", "car", "chair",
"chandelier", "clothes dryer", "coffee maker", "column",
"commode", "couch", "counter", "countertop", "crib",
"desk", "dishwasher", "door", "drawer", "dresser",
"fireplace", "jalousie", "microwave", "oven", "pillar",
"pool table", "radiator", "range hood", "refrigerator",
"screen", "shelf", "stand", "stool", "stove", "table",
"toilet", "tv", "vanity", "wardrobe", "washing machine",
"window"]
```

We perform unconditional scene generation experiments using ATISS [23], DiffuScene [29], and PhyScene [36]. The implementations of these methods are adapted from their official GitHub repositories to fit our dataset. For the two diffusion-based methods, DiffuScene and PyScene, we set the maximum number of objects per scene to 50. To ensure fair comparison across methods, all baselines adopt the same network architecture, training hyperparameters, and experimental setup. In addition, the object retrieval process for constructing 3D scenes and the rendering pipeline used for metric computation are kept identical.

**Qualitative Results and Analysis.** We present the results of unconditional scene generation using the three baseline methods on both the simplified version and full version datasets in Figure12 and Figure13, respectively. By comparing the generation results, we observe that baseline models trained on the full version of the dataset tend to produce erroneous layouts for small objects, such as floating or interpenetrating artifacts. These models struggle to accurately control the position and orientation of small objects to ensure physical plausibility. In addition, there are qualitative differences in the placement of large furniture between the two versions of InternScenes. Scenes generated using the simplified version exhibit more reasonable layouts for large objects compared to those generated from the full version. This may be caused by the limited contextual modeling capacity of existing baseline models when handling scenes with a large number of objects, making it difficult to effectively capture the layout distribution in the InternScenes dataset. A new challenge of scene generation is to enable models to better learn the layout distribution of complex scenes containing numerous objects and to generate scenes that are more physically realistic.

## B.2 Navigation

Examples of the evaluation scenes for navigation are visualized in Figure 14. We bind the collider for all the meshes in the scenes and download the robot asset of ClearPath Dingo from the official Isaacsim assets as the navigation robot. To decide the starting points and target points for each evaluation episode, we extract the floor as the navigable areas and calculate the ESDF map. The navigable areas with ESDF value greater than $0.5$m are filtered as candidates. Finally, we randomly sample pairs of points with distances in the range $(3\text{m}, 10\text{m})$ as the starting and destination for navigation. For a physical-realistic evaluation benchmark, we control two wheel speeds for Dingo in the IsaacSim, instead of teleporting the agent to the predicted pose of the navigation methods. To decide the wheel speed, we first convert the baseline navigation methods' prediction results into linear and angular speed, then calculate the desired wheel speed with a differential model. For the DD-PPO, as this method is trained with discrete action space and predicts among four actions $\{MoveForward, TurnLeft, TurnRight, Stop\}$, we simply map each discrete action into a pre-defined speed set $\{(u = 0.5, w = 0.0), (u = 0.0, w = 1.0), (u = 0.0, w = -1.0), (u = 0.0, v = 0.0)\}$, where $u$ represents the linear speed and $v$ represents the angular speed. For the NavDP, as this method predicts a continuous trajectory, we select the fourth waypoint in the trajectory and convert the waypoint coordinates into linear and angular speed by an open-loop controller. The linear speed is calculated with a coefficient $K_u$ multiplying the L2-norm of the waypoint coordinates, and the angular speed is calculated with a coefficient $K_w$ multiplying the relative yaw angle between the fourth waypoint and the current pose.

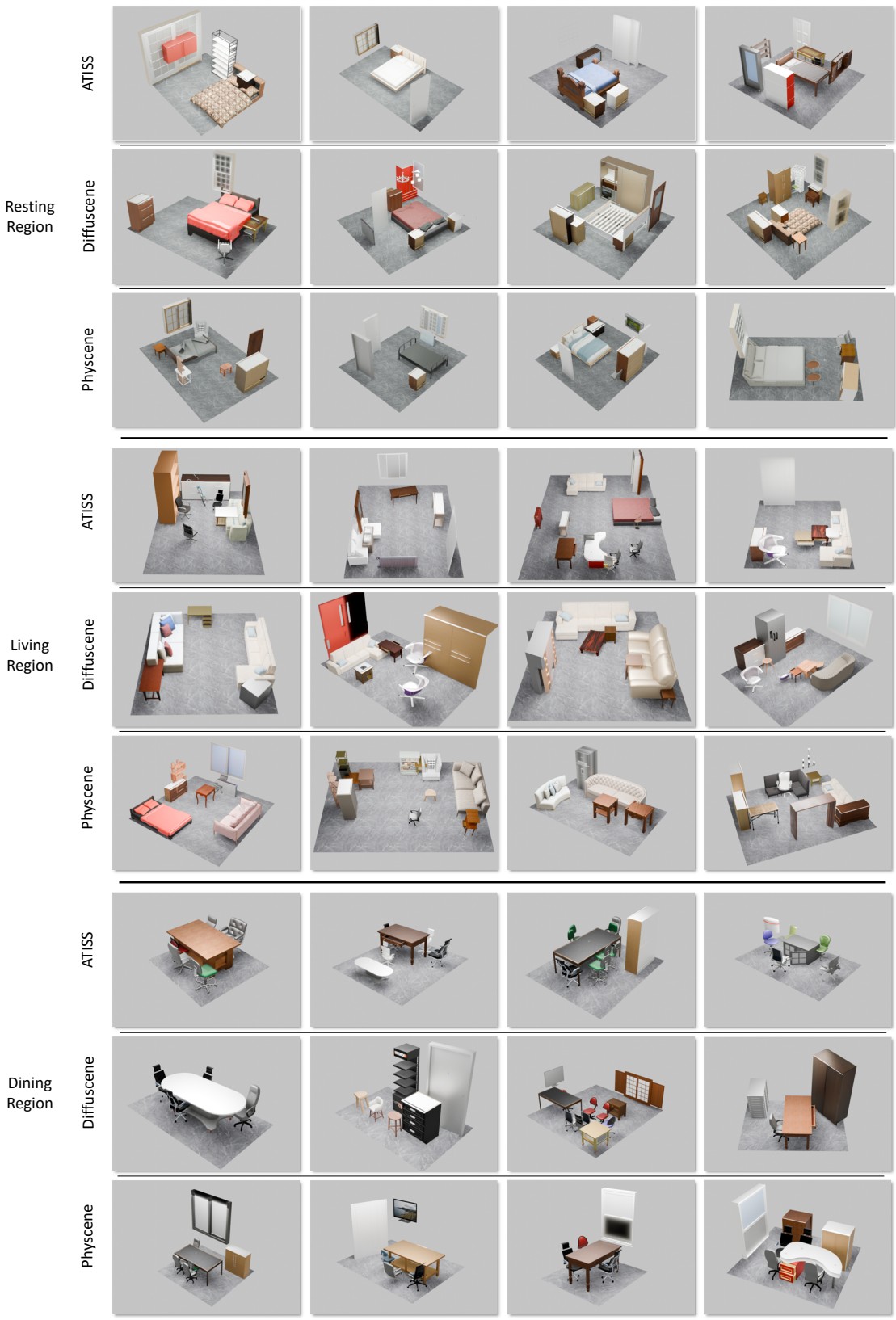

Figure 12: Examples of regions generated by baseline models trained on a simplified version of the InternScenes dataset

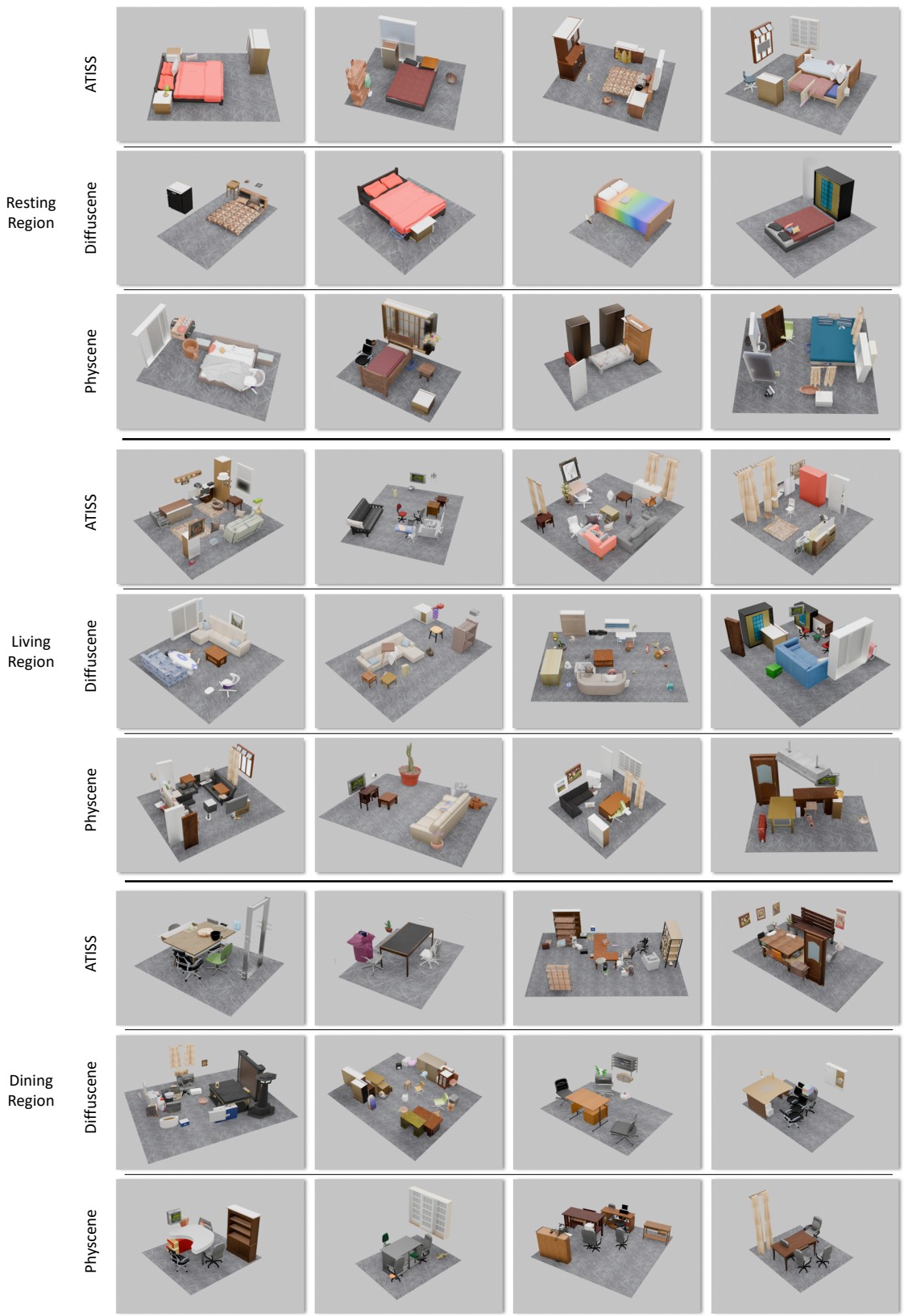

Figure 13: Examples of regions generated by baseline models trained on the full version of the InternScenes dataset

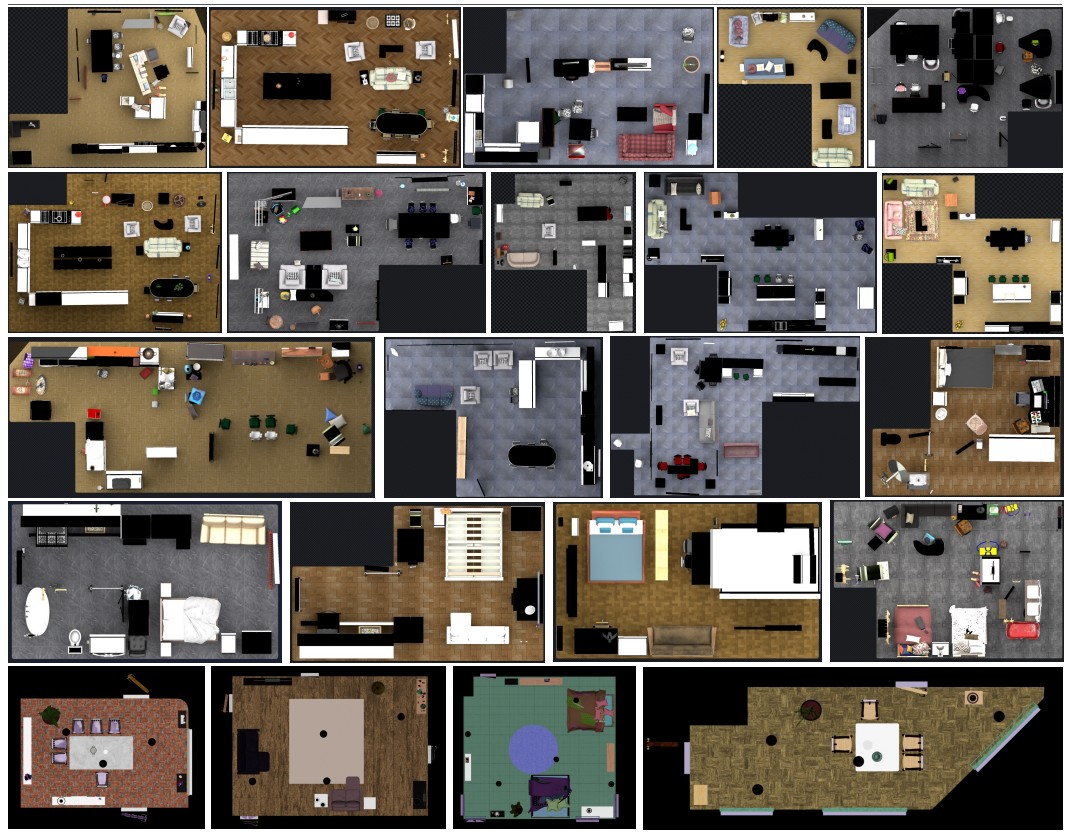

Figure 14: Scenes for the navigation evaluation.

## C Dataset Statistics

**Scene Showcase.** We provide some scene examples of InternScenes for visualization. Figure 15 shows some examples of *InternScenes-Real2Sim*, where each scene originates from a scanned real-world room and is then transformed via a real-to-sim transformation. In Figure 16, we show some examples of *InternScenes-Gen*, which are constructed using procedural generation techniques. Moreover, Figure 17 showcases curated scenes created by professional designers from *InternScenes-Synthetic*.

**Layout and Objects Statistics.** Our dataset comprises three subsets, totaling 39870 scenes and 48381 regions across 15 categories. Specifically, the InternScenes-Real2Sim subset contains 9833 regions, InternScenes-Gen contains 11454 regions, and InternScenes-Synthetic contains 27094 regions. In total, 1.96M objects from 288 categories are placed across all regions, sampled from our asset library of 80M CAD models. These objects are sampled from our asset library containing 80 million CAD models. On average, each region contains 41.5 objects. We also conduct a statistical study of the volume distribution of all object bounding boxes in the scenes (Figure 18(a)), and further analyzed the volume distributions of five representative object categories. These categories were selected to represent objects of varying scales, ranging from large furniture to small items: *chair*, *bed*, *couch*, *bottle*, and *book* (Figure 18(b)).

**Data Format.** The dataset is structured into two primary components: region-level layout information and a model asset library. The layout information is characterized by the semantic attributes of each region and the objects it contains. For each region, detailed object annotations are provided, including the corresponding model name from the asset library, object category, spatial center coordinates, bounding box dimensions, and associated $ZXY$ Euler angles. The model asset library contains mesh representations of all objects, enabling complete 3D scene reconstruction when combined with the layout information.

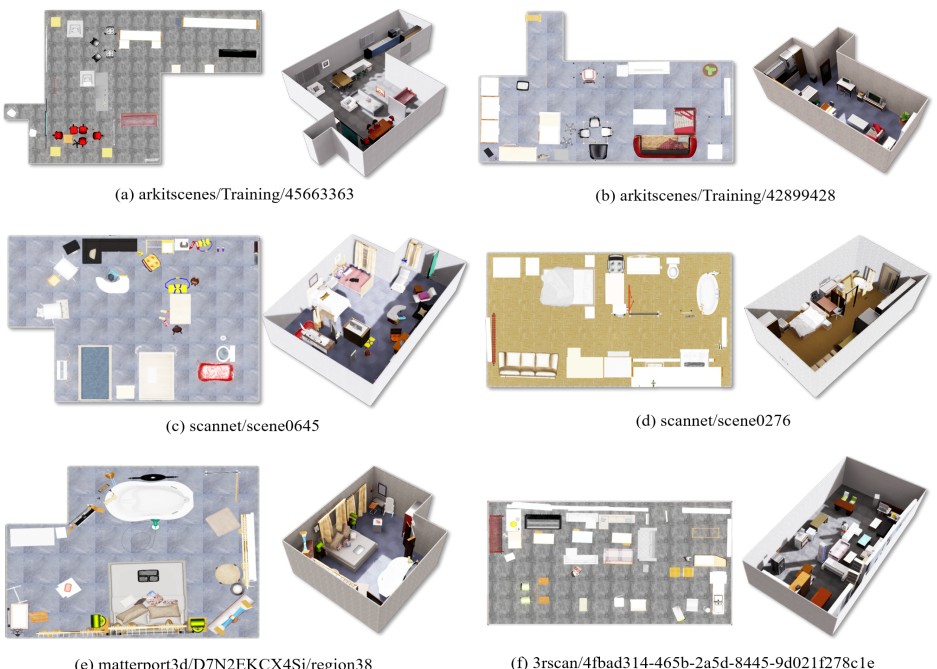

(a) arkitscenes/Training/45663363

(b) arkitscenes/Training/42899428

(c) scannet/scene0645

(d) scannet/scene0276

(e) matterport3d/D7N2EKCX4Sj/region38

(f) 3rscan/4fbad314-465b-2a5d-8445-9d021f278c1e

Figure 15: Examples from InternScenes-Real2Sim. Each scene shows its BEV map as well as one isometric view.

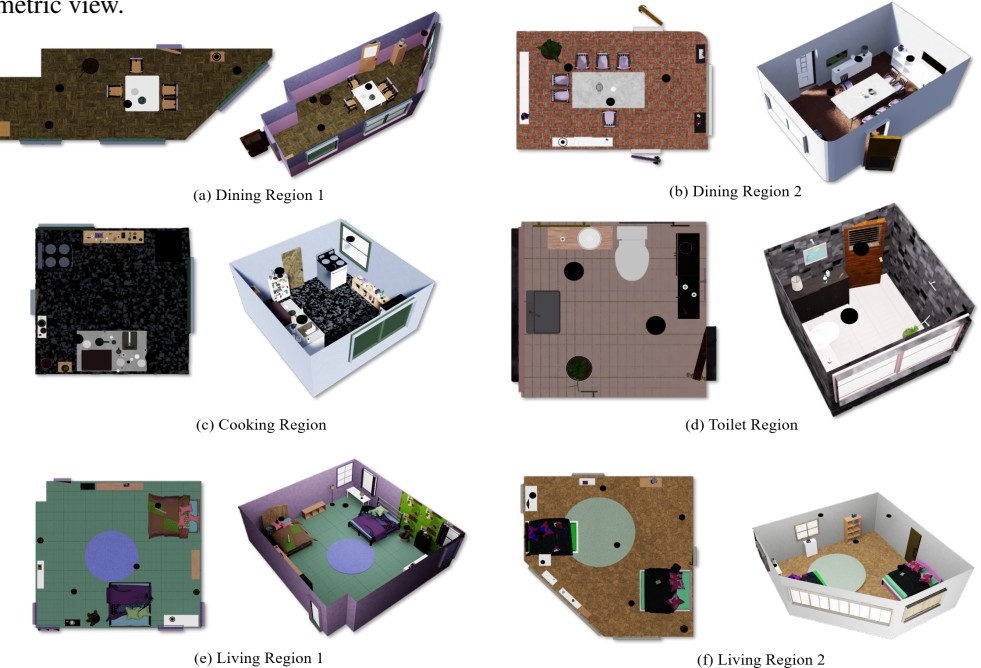

(a) Dining Region 1

(b) Dining Region 2

(c) Cooking Region

(d) Toilet Region

(e) Living Region 1

(f) Living Region 2

Figure 16: Examples from InternScenes-Gen. The BEV map and one isometric view are shown.

**Region and Object Joint Statistics.** Figure 19 illustrates the distribution of object density, measured as the number of objects per square meter ($m^2$), across various regions. The average object density computed across all scenes is 1.296 objects per square meter.

In Figure 20, we show the distribution of 100 object categories across 15 regions, where the depth of the rectangle's color and its size are positively correlated with the quantity of that object category within the corresponding region. The darker and larger the rectangle, the higher the frequency of that object category in the area.

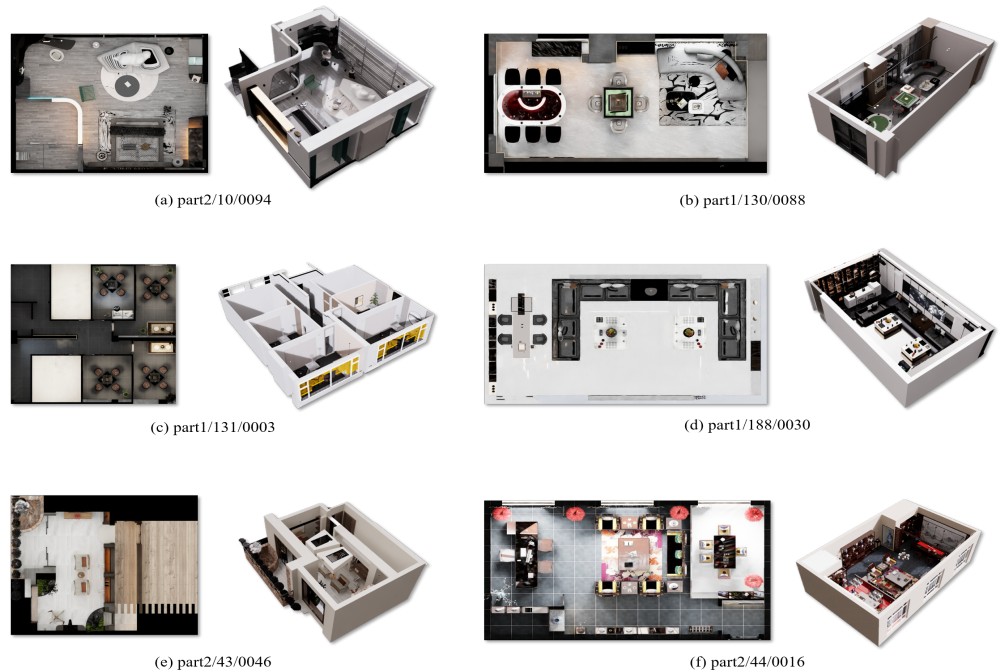

(a) part2/10/0094       (b) part1/130/0088

(c) part1/131/0003       (d) part1/188/0030

(e) part2/43/0046       (f) part2/44/0016

Figure 17: Examples from InternScenes-Synthetic. The BEV map and one isometric view are shown.

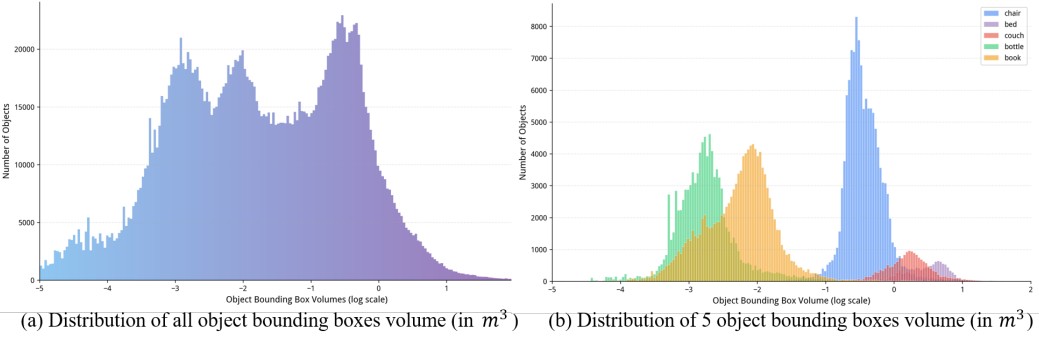

(a) Distribution of all object bounding boxes volume (in $m^3$)     (b) Distribution of 5 object bounding boxes volume (in $m^3$)

Figure 18: Object bounding boxes volume statistics

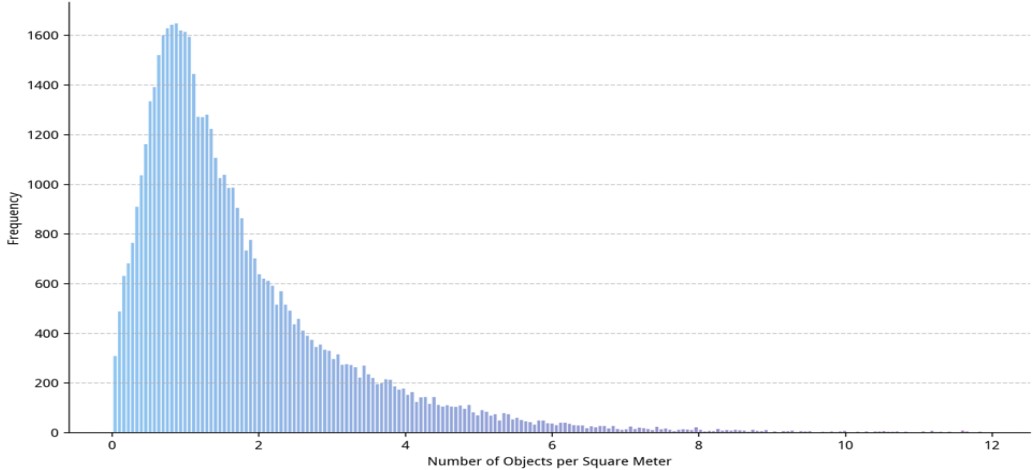

Figure 19: Distribution of object density (number of objects per $m^2$) across different regions.

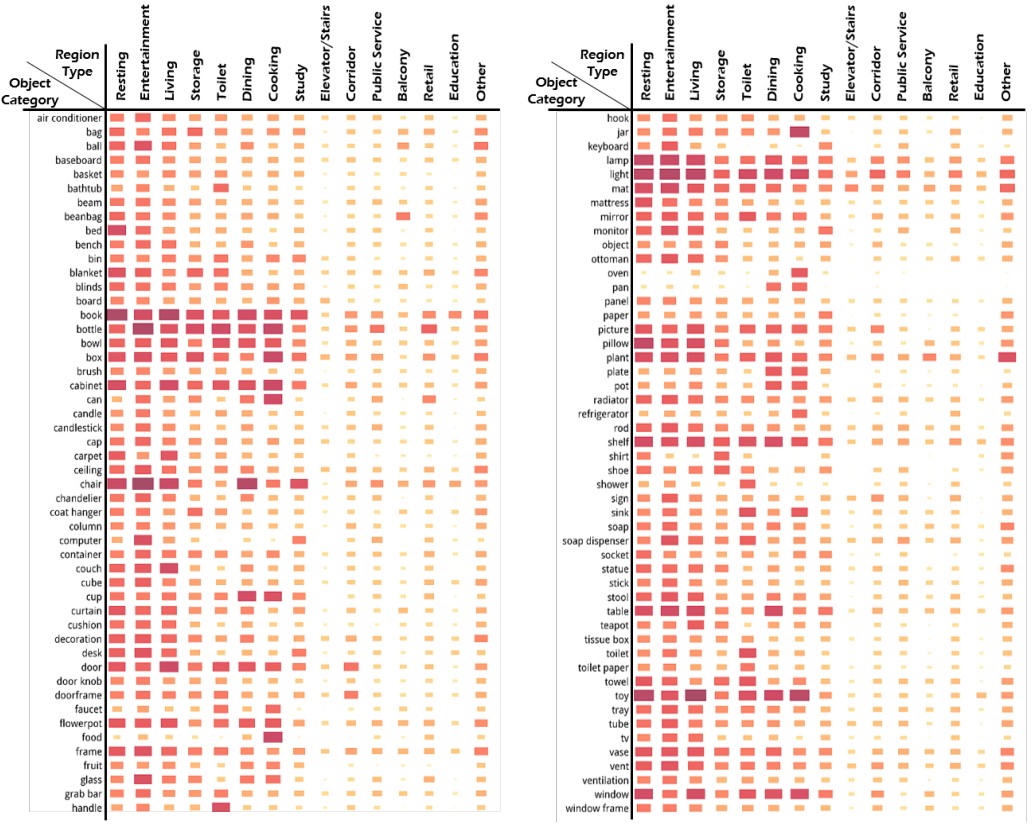

Figure 20: Distribution of 100 object categories conditioned on 15 different types

Table 6: FPS results (min–max / mean) under different levels of parallel simulation.

| Scene Type | Parallel=1 | Parallel=20 | Parallel=40 |
|---|---|---|---|
| OmniScenes-Real2Sim | 242.86-250.48 / 246.95 | 173.39-177.83 / 175.34 | 127.77-136.44 / 131.86 |
| OmniScenes-Gen | 242.29-275.43 / 263.95 | 225.19-244.65 / 238.22 | 141.28-212.31 / 200.07 |
| OmniScenes-Synthetic | 172.06-176.83 / 175.04 | 84.41-88.29 / 86.57 | 50.74-52.06 / 51.61 |

Table 7: CPU usage (cores % / memory in GB) under different levels of parallel simulation.

| Scene Type | Parallel=1 | Parallel=20 | Parallel=40 |
|---|---|---|---|
| OmniScenes-Real2Sim | 4.588% / 5.602 GB | 9.924% / 10.649 GB | 12.051% / 19.606 GB |
| OmniScenes-Gen | 16.129% / 4.938 GB | 14.480% / 7.927 GB | 21.320% / 12.629 GB |
| OmniScenes-Synthetic | 73.623% / 13.879 GB | 79.718% / 21.923 GB | 73.400% / 32.388 GB |

Table 8: GPU memory usage (in GB) under different levels of parallel simulation.

| Scene Type | Parallel=1 | Parallel=20 | Parallel=40 |
|---|---|---|---|
| OmniScenes-Real2Sim | 2.528 GB | 5.205 GB | 5.385 GB |
| OmniScenes-Gen | 5.399 GB | 5.476 GB | 5.679 GB |
| OmniScenes-Synthetic | 7.542 GB | 7.785 GB | 8.168 GB |

# D System Performance and Resource Requirements

**Detailed Performance Metrics.** We perform a comprehensive evaluation of the resource overhead associated with rendering our scenes in the simulator. Specifically, we report the following metrics after rendering scenes from three subsets of our dataset in Isaac Sim: GPU memory usage, CPU usage and memory usage (%), and rendering throughput (FPS) All experiments are conducted on a high-performance node equipped with 128 vCPUs, 1024 GB DDR5 RAM, and 8× NVIDIA RTX 4090 (48 GB) GPUs. Isaac Sim is launched in headless mode using 1 GPU, 16 CPU cores, and 128 GB RAM. We adopt the *Stage Light* environment for for all scenes. After loading a scene, we run the simulator for 2000 steps, discarded the first 200 steps as warm-up, and computed FPS and memory footprints on the remaining steps.

The detailed results are summarized in Table 6, 7, 8. These metrics provide a clear view of the computational cost and rendering efficiency of our dataset under realistic simulation conditions.

**Parallel Simulation Support.** To evaluate the dataset's support for parallel simulation, we conduct experiments under the same hardware and runtime configuration described before. Specifically, we select scenes from each of the three data sources in our dataset.

We then incrementally load multiple scenes into a single Isaac Sim *World* in headless mode, and monitored the system performance as the number of parallel environments increased. The quantitative results are reported in Table 6, 7, 8.

# E Discussion on Procedural Generation with Infinigen Indoor

To enrich the diversity of generated assets and layouts in our dataset, we leverage Infinigen Indoors [24], a procedural generation framework designed to mitigate risks of introducing bias in spatial configurations and object co-occurrence patterns through fully randomized asset generation and a constraint-based layout optimization that utilizes simulated annealing, thereby minimizing systematic bias.

For the assets in the omniscenes-gen of our dataset, Infinigen Indoors generates objects with extensive randomization. For example, the furniture category alone includes 17 generators with 216 controllable parameters. This high degree of parameterization ensures significant diversity in the generated assets, which in turn avoids the explicit bias introduced by reusing a static set of models.

Regarding spatial configurations, Infinigen Indoors employs a Simulated Annealing solver to search a large state space for generating the scenes, which prevents the inclusion of templated or repetitive layouts in our data. According to the Infinigen Indoors [24], the Simulated Annealing solver uses the following pipeline to ensure the diversity and randomness of the generated scenes:

- At each iteration, given the current scene state $s$, the solver randomly chooses a move to apply to the scene (e.g., adding or rotating an object), generating a proposed state $s'$.

- Both the original state $s$ and the proposed state $s'$ are evaluated on the constraint graph, yielding corresponding loss terms $l(s)$ and $l(s')$. The probability of accepting the new state is given by:

$$p(s'|s) = \min\left[\exp\left(\frac{l(s) - l(s')}{\tau}\right), 1\right]$$

where $\tau$ is the current temperature. This indicates that if the new scene state is an improvement, it is always accepted. However, if the new layout is not an improvement, the solver may still accept it with a certain probability.

- As the optimization progresses, the temperature parameter $\tau$ cools from $0.25$ to $0.001$.. This means that in the initial phase of optimization, the solver has a higher probability of accepting a state with a higher loss, allowing it to escape local optima and perform a broader exploration of the solution space. In the final stages, the solver almost exclusively accepts better states, allowing the scene to converge to a high-quality arrangement.

This optimization mechanism employed by Infinigen Indoors ensures that our scenes have diverse and high-quality spatial configurations, thereby suppressing the generation of bias.

# F  Support for Articulated Objects

**Articulated Objects Source.**  We use URDF assets from the PartNet-Mobility dataset, a peer-reviewed and widely used authoritative dataset for robot manipulation research, where the quality and annotation accuracy of its URDF files have been validated by the community.

**Integration Pipeline.**  A programmatic pipeline assembles these assets into interactive scenes within Isaac Sim, ensuring that the kinematic structure defined in the URDF files is faithfully preserved as articulations in the final USD scene. The pseudocode for assembling an interactive scene in Isaac Sim is as follows:

---

**Pseudocode for Assembling an Interactive Scene**

```python
# Input: layout_file (our provided JSON), asset_path (path to
    assets)

def build_interactive_scene(layout_file, asset_path):
    # 1. Initialize a new USD stage in Isaac Sim
    stage = IsaacSim.create_new_stage()
    layout_data = parse_json(layout_file)

    # 2. Iterate through objects defined in our layout file
    for obj_info in layout_data["objects"]:
        prim_path = f"/World/{obj_info.name}"
        full_asset_path = asset_path + obj_info.asset_file

        # 3. Conditionally import assets based on type
        if obj_info.type == "glb":  # For static objects
            stage.add_reference_to_prim(prim_path,
                full_asset_path)
        elif obj_info.type == "urdf":  # For articulated objects
            # Use Isaac Sim's standard URDF importer
            # This automatically creates a physics-enabled
                articulation
            IsaacSim.URDF_Importer.import(
                urdf_path=full_asset_path,
                prim_path=prim_path,
                create_articulation=True  # This is the key for
                    interactivity
            )

        # 4. Set the object's pose in the world
        prim = stage.get_prim(prim_path)
        prim.set_world_transform(obj_info.position, obj_info.
            orientation)

    return stage
```

---

This pipeline guarantees that once an object is imported, its joints (e.g., a cabinet's hinge) are not just visual elements but are fully interactive, physics-driven articulations ready for manipulation tasks. In addition, every step of this process relies on Isaac Sim's core, publicly documented APIs, such as the `omni.importer.urdf` tool. This proves that our dataset is not only theoretically usable but also practically direct and convenient to use, as it seamlessly integrates into the standard Isaac Sim workflow.

