# OpenReview forum: "InternScenes: A Large-scale Simulatable Indoor Scene Dataset with Realistic Layouts"
_NeurIPS.cc/2025/Datasets_and_Benchmarks_Track — NeurIPS 2025 Datasets and Benchmarks Track poster_

### Official Review · Reviewer_JbmP · 2025-06-26

**Rating:** 4
**Confidence:** 4

**Summary:**

The paper presents OmniScenes, a large-scale indoor scene dataset comprising 40K scenes from three data sources (1) real-world scans (EmbodiedScan), (2) procedurally generated layouts (Infinigen Indoors), and (3) designer-created scenes. The paper applies a multi-stage pipeline for real-to-sim conversion, annotation, and physics-based collision resolution. The authors benchmark OmniScenes on two tasks: interior scene layout generation (full vs. simplified versions) and point-goal visual navigation, demonstrating novel challenges posed by dense, realistic layouts.

**Dataset Code Accessibility:**

Partly

**Dataset Code Comments:**

While scene region information is uploaded to HuggingFace, the orientation-aligned mesh objects are not yet available, which limits the dataset’s usability. This is understandable, as mesh files can be large, but the authors should clarify how to process or convert raw data from the Objaverse and PartNet-Mobility datasets into the required format.
The code for curating the Real2Sim and synthetic subsets has been released.

**Ethical Comments:**

The dataset involves real2sim or synthetic 3D scenes without direct ethic problems.

**Ethical Considerations:**

No, there are no or only very minor ethics concerns

**Final Justification:**

The rebuttal adequately addresses my concerns through detailed ablations and clarifications regarding dataset availability, and I will raise my score to borderline accept accordingly.

**Limitations Weaknesses:**

- Given that the dataset integrates three different sources, a more comprehensive discussion on the quality and characteristics of each source would be valuable. For instance, OmniScenes-Gen can potentially generate infinite samples—does this abundance degrade performance on downstream tasks due to distribution mismatch or overfitting? Meanwhile, OmniScenes-Real2Sim contains only 9K samples; it would be helpful to analyze the impact of incorporating this realistic data source.
- The dataset is not immediately ready for use, and the paper lacks sufficient details on the processing of mesh objects, making it difficult to use or reproduce.

**Strengths Contributions:**

- The proposed OmniScenes dataset combines multiple data sources to curate a unified 3D indoor scene dataset with rich annotations.
- The dataset statistics demonstrate enhanced diversity and quality compared to existing benchmarks.
- Two tasks are selected to showcase the benefits and practical usage of the dataset.

---

> ### Author Rebuttal · Authors · 2025-07-31
>
> We sincerely thank you for your recognition of our work on the dataset and for your valuable feedback. We are pleased that you acknowledge the contributions of OmniScenes in integrating multiple data sources, enhancing the dataset's diversity and quality, and demonstrating its application value through two downstream tasks. Regarding the two limitations you have raised, we will address them in detail below.
>
> ## W1: Analysis of Different Data Composition Ratios
>
> We greatly appreciate your raising this insightful question. Our core design philosophy is that no single source can simultaneously meet all the requirements of large-scale, layout realism, and diversity. OmniScenes, by integrating the strengths of three sources, forms a complementary and more robust dataset.
>
> The unique contribution of each data source:
>
> - OmniScenes-Real2Sim (from EmbodiedScan): This subset, derived from real-world scanned scenes, features highly realistic object layouts. Although the scale (9,833 regions ) is smaller than the others, it includes a large number of small objects that are often omitted in previous non-OmniScenes datasets, significantly enhancing the realism of the indoor scene layouts.
>
> - OmniScenes-Gen (generated from Infinigen Indoors): OmniScenes-Gen consists of 11,454 regions, covering 5 common scene categories. We utilize the rule-based generator from Infinigen Indoors to create scenes with high physical plausibility and zero-collision properties.
>
> - OmniScenes-Synthetic (from Designer-Created Scenes): This subset, with 27,094 regions, is the primary contributor to the dataset's overall scale and diversity. It offers the broadest spatial coverage and a diverse combination of functional areas, compensating for the limitations of scanned data in terms of scene types and scale.
>
> To provide a quantitative answer to your question, we conducted a new set of scene layout generation experiments to analyze how varying the proportions of our data sources impacts downstream task performance.
>
> - **Method**: We leverage PhyScene [1], a strong and representative scene generation method, to conduct the ablation study.
>
> - **Varied data mixture proportions**: We create three distinct training sets with different data composition ratios across three typical region types, the Resting(4,620 regions), Living(6,181 regions), and Dining (1,883 regions). In each training set, the large-scale OmniScenes-Synthetic subset is kept constant, comprising a sample count 1.5 times larger than the OmniScenes-Real2Sim and OmniScenes-Gen subsets combined. We then vary the ratio of these two specialized sources at 3:1, 2:2, and 1:3 to isolate their respective influences. To rigorously evaluate generalization and avoid testing on a distribution identical to the training data, we used the 3D-FRONT [2] dataset as an external test set. This demonstrates how well models trained on OmniScenes perform on out-of-distribution data.
>
> - **Metrics**: We used standard scene generation metrics: FID (↓), CKL (↓), and SCA (closer to 50% is better).
> The quantitative results of our experiments are presented in the table below:
>
> | RegionType | Real2Sim:Gen Ratio | FID (↓) | CKL (x0.01) (↓) | SCA (%) |
> | :---: | :---: | :---: | :---: | :---: |
> |  | 3 : 1 | 70.24 | 4.01 | 99.20 |
> | **Resting Region** | 2 : 2 | 63.80 | 3.27 | 98.16 |
> | | 1 : 3 | 82.52 | 5.76 | 99.18 |
> |  | 3 : 1 | 89.76 | 9.56 | 99.54 |
> |**Living Region** | 2 : 2 | 80.61 | 7.76 | 99.85 |
> |  | 1 : 3 | 93.52 | 8.94 | 99.69 |
> |  | 3 : 1 | 83.72 | 9.56 | 99.23 |
> | **Dining Region** | 2 : 2 | 81.16 | 8.77 | 99.57 |
> |  | 1 : 3 | 89.52 | 10.79 | 99.21 |
>
> Across all three region types, the balanced 2:2 ratio consistently achieves the best performance. This demonstrates the model generalizes best when it learns from both Real2Sim and  OmniScenes-Gen.
>
> The reviewer rightly questioned whether the procedural nature of OmniScenes-Gen could lead to a distribution mismatch. Our results for the 1:3 ratio (more OmniScenes-Gen scenes in the training set) confirm this is a valid concern. In this setting, performance degrades (e.g., FID for the Resting region increases from 70.24 to 82.52). While OmniScenes-Gen provides clean and diverse layouts, over-relying on it can cause the model to struggle when generalizing to the priors of an external dataset like 3D-FRONT. This explains our design choice to balance the contribution of OmniScenes-Gen to be comparable with OmniScenes-Real2Sim, rather than allowing it to dominate the dataset's composition.
>
> The 3:1 ratio, which emphasizes the realistic OmniScenes-Real2Sim data, performs better than the 1:3 ratio but falls short of the balanced 2:2 mix. This answers the reviewer's second question: incorporating the 9.8K realistic samples from OmniScenes-Real2Sim is crucial to improving the model's ability to generate scenes with realistic layout.
>
> In summary, these experiments quantitatively validate our approach. OmniScenes forms a more robust dataset by integrating the strengths of three sources. This composition mitigates the weaknesses of each individual source, resulting in a more robust and powerful dataset for training generalizable models.
>
> ---
> ## W2: Clarification on Dataset Usability and Mesh Processing Pipeline
>
> We sincerely apologize for the confusion caused by the lack of clear documentation in the original GitHub and Hugging Face repositories at the time of submission. We appreciate the reviewer’s feedback and would like to provide a more detailed explanation of both the dataset structure and our mesh object processing pipeline.
>
> ### 1. Dataset Availability and Structure
>
> The dataset shared via the Hugging Face link includes a `layout.json` file, which specifies the scene composition using object-level information. Each object is identified by a **PartNet-Mobility** ID and a corresponding **Objaverse** UID. Using these identifiers, users can retrieve the original 3D assets and their bounding box placements to reconstruct the full scene.
>
> However, we acknowledge that we did not provide a convenient script or detailed instructions to automatically download the assets and compose the scenes. This may have made the dataset appear less accessible and harder to use, and we apologize for the inconvenience.
>
> ### 2. Mesh Object Processing Pipeline
>
> We elaborate here on how object meshes are processed, as illustrated in Figures 2 and 3 of our main paper.
>
> - For **OmniScenes-Real2Sim** scenes, we employed a combination of **GPT-based semantic annotation** and a publicly available **Vision-Language Model (VLM)** to estimate the canonical orientation (i.e., the front-facing direction) of each object. All annotations were manually verified to ensure correctness.
> - For **OmniScenes-Synthetic** scenes, we built an internal annotation tool to assist in mesh segmentation and merging. Through this tool, we achieved an annotation accuracy of over **95%**, ensuring high-quality object-level mesh assets.
>
> Once the processed object meshes are prepared, we place them into the scene using the bounding box information provided in the `layout.json` files. This process allows us to compose the full 3D scene accurately and consistently.
>
> ### 3. Future Updates and Reproducibility
>
> We have already prepared the asset library and supporting scripts necessary to make the dataset fully ready-to-use. However, due to **NeurIPS rebuttal policy restrictions**, we are unable to modify the repositories during the review period. We commit to updating the repositories immediately after the rebuttal phase, including:
>
> - Scripts for automated asset downloading and scene composition
> - Detailed documentation of the mesh processing
> - The mesh assets we have processed
>
> We hope this clarification addresses the reviewer’s concern and highlights our commitment to **open and reproducible research**.
>
> [1]PhyScene: Physically Interactable 3D Scene Synthesis for Embodied AI, CVPR 2024
>
> [2]3D-FRONT: 3D Furnished Rooms with layOuts and semaNTics, CVPR 2021

---

> > ### Comment · Reviewer_JbmP · 2025-08-02
> >
> > Thank you to the authors for the rebuttal. It adequately addresses my concerns through detailed ablations and clarifications regarding dataset availability. I will raise my score accordingly.

---

> > > ### Author Response · Authors · 2025-08-04
> > >
> > > We appreciate your acknowledgement. We will include these ablations in the final version and provide clarifications on Hugging Face to facilitate dataset accessibility.

---

### Official Review · Reviewer_UMom · 2025-06-30

**Rating:** 5
**Confidence:** 4

**Summary:**

This paper introduces OmniScenes, a large-scale and simulatable 3D indoor scene dataset that integrates real-world scans, generated layouts and designer-created scenes. It addresses key limitations in existing datasets, including lack of small objects and object collisions. OmniScenes supports benchmarks in scene layout generation and point-goal navigation, both of which reveal new challenges for current models in complex environments.

**Dataset Code Accessibility:**

Yes

**Ethical Considerations:**

No, there are no or only very minor ethics concerns

**Final Justification:**

The authors’ rebuttal has satisfactorily addressed my concerns.

**Limitations Weaknesses:**

1. The current benchmarks focus only on scene generation and point-goal navigation, while other embodied AI tasks such as manipulation, object search or vision-language grounding are not yet explored.
2. The procedural scenes are derived solely from Infinigen Indoors, which, while rule-rich, may introduce bias in spatial configurations and object co-occurrence patterns.

**Strengths Contributions:**

1. The dataset features high object density (41.5 objects per region) and includes small and interactive items, which are often omitted in other datasets.
2. By resolving object collisions and generating physically plausible layouts, OmniScenes is natively compatible with simulators like IsaacSim.
3. Both scene generation and navigation baselines struggle on OmniScenes, demonstrating its effectiveness as a stress test for future models.

---

> ### Author Rebuttal · Authors · 2025-07-31
>
> We sincerely thank you for your thorough evaluation and constructive feedback. We are especially grateful for your recognition of OmniScenes' key strengths, including its high object density with small and interactive items often omitted in other datasets , its native simulatability achieved by resolving object collisions for use in engines like IsaacSim , and its demonstrated value as a 'stress test' for future models. In the following, we will address your valuable concerns.
>
> ---
>
> ## W1: Visual Grounding Experiments on our Dataset
> We thank the reviewer for the insightful suggestion. While our main benchmarks focus on scene generation and point-goal navigation, we have also explored the potential of our dataset in supporting vision-language grounding tasks. Specifically, we conducted experiments on the visual grounding task using the **ScanRefer** model.
>
> ### Data Preparation
>
> To enable this, we leveraged the existing annotated data from the **EmbodiedScan** dataset (referred to as the real scan), and constructed a corresponding Omniscenes-Real2Sim version using our retrieval pipeline. In this retrieval version:
>
> - The original real-scanned point clouds were replaced with point clouds rendered from CAD models.
> - The original bounding boxes were substituted with those generated from simulation.
>
> This approach significantly reduces the time and effort required to create such data from scratch. Furthermore, the same data generation pipeline can be readily adapted to other data sources in OmniScenes, enabling the efficient construction of visual grounding datasets in a manner similar to how EmbodiedScan data was utilized.
>
> ### Experiment Setup
>
> We used the **ScanRefer** model as a baseline and conducted two additional experiments:
>
> 1. **Mixed Training and Real Testing**:
>    Trained the ScanRefer model on a combination of real scan data and its corresponding retrieval version, and tested on real scan data.
>
> 2. **Retrieval-Only Training and Testing**:
>    Trained and tested the model purely on the retrieval data.
>
> ### Results
>
> The results are summarized below (measured by **AP@0.25**):
>
>
> |   | Training  | Test  | AP 25 |
> |---|-------------------|--------------|-------|
> | 1 | Real              | Real         | 12.85 |
> | 2 | Real  + Retrieval | Real         | 15.18 |
> | 3 | Retrieval         | Retrieval    | 16.49 |
>
> ### Analysis
>
> Comparing **Experiment 1** (training on real data only) and **Experiment 2** (training on real + retrieval data), we observe a performance improvement, indicating that the retrieved scenes serve as effective data augmentation for the original real scenes.
>
> Furthermore, the comparison between **Experiment 1** and **Experiment 3** (training and testing on retrieval data) shows that our retrieval pipeline preserves the layout distribution of the original scenes.
>
> These findings suggest that our **Omniscenes-Real2Sim** data is both **realistic** and **complementary** to real-world data.
>
> ---
>
> ## W2: Discussion on Procedural Generation with Infinigen Indoors
>
> We thank the reviewer for their insightful comment regarding the potential for procedural generation to introduce bias in spatial configurations and object co-occurrence patterns. We acknowledge this is a critical challenge for any generative model. We chose Infinigen Indoors precisely because it is designed to directly mitigate these specific risks through fully randomized asset generation and a constraint-based layout optimization that utilizes simulated annealing, thereby minimizing systematic bias.
>
> For the assets in the omniscenes-gen of our dataset, Infinigen Indoors generates objects with extensive randomization. For example, the furniture category alone includes 17 generators with 216 controllable parameters. This high degree of parameterization ensures significant diversity in the generated assets, which in turn avoids the explicit bias introduced by reusing a static set of models.
>
> Regarding spatial configurations, Infinigen Indoors employs a Simulated Annealing solver to search a large state space for generating the scenes, which prevents the inclusion of templated or repetitive layouts in our data. According to the Infinigen Indoors [1] paper, the Simulated Annealing solver uses the following pipeline to ensure the diversity and randomness of the generated scenes:
>
> - At each iteration, given the current scene state $s$, the solver first randomly chooses a move to apply to the scene (e.g., adding or rotating an object) to generate a proposed state $s'$.
>
> - Both the original state $s$ and the proposed state $s'$ are then evaluated on the constraint graph, yielding corresponding loss terms, $l(s)$ and $l(s')$. The probability of accepting the new state is given by:  $p(s^{\prime}|s)=\min\left[\exp\left(\frac{l(s)-l(s^{\prime})}{\tau}\right),1\right]$. This indicates that if the new scene state is an improvement, it is always accepted. However, if the new layout is not an improvement, the solver may still accept it with a certain probability.
>
> - As the optimization progresses, the temperature parameter $\tau$ cools from 0.25 to 0.001. This means that in the initial phase of optimization, the solver has a higher probability of accepting a state with a higher loss, allowing it to escape local optima and perform a broader exploration of the solution space. In the final stages, the solver almost exclusively accepts better states, allowing the scene to converge to a high-quality arrangement.
>
> In summary, this optimization mechanism ensures that our scenes have diverse and high-quality spatial configurations, thereby suppressing the generation of bias. We again thank the reviewer for this valuable feedback. We hope this clarification of the Infinigen method effectively demonstrates the soundness of our choice and resolves the raised concern.
>
> [1] Infinigen indoors: Photorealistic indoor scenes using procedural generation, CVPR 2024

---

> > ### Comment · Reviewer_UMom · 2025-08-09
> >
> > The authors’ rebuttal has addressed my concerns, and I will accordingly raise my score.

---

> ### Author Response · Authors · 2025-08-07
>
> Dear Reviewer,
>
> Thank you once again for reviewing our paper. We would greatly appreciate it if you could take a moment to review our feedback, and let us know if any concerns remain.
>
> Best regards,
>
> The Authors

---

### Official Review · Reviewer_QDj9 · 2025-07-03

**Rating:** 4
**Confidence:** 3

**Summary:**

This paper introduces a large-scale indoor scene dataset named OmniScenes. OmniScenes contains three types of scenes, namely, synthetic scenes with procedural generation, synthetic scenes created from expert designers, and real-world scenes. Especially, for the last type, the authors propose a method to convert real-world scenes into simulation-ready replicas. Also the authors show the experimental results on scene generation task and point goal navigation task. The experiments show that OmniScenes opens new challenges in the indoor scene understanding field.

**Additional Feedback:**

Overall, I appreciate the authors for creating large-scale simulatable indoor scene dataset, and I believe this dataset can be serve as a valuable asset for robotics and indoor scene understanding fields. However, my major concern is accessibility to the dataset and effectiveness of articulated objects.

**Dataset Code Accessibility:**

Partly

**Dataset Code Comments:**

There exists code repository (https://github.com/PC1E-bit/OmniScenes) to create the dataset. However, there is no document for the code at all. Also, dataset is not publicly available yet, which is important for the dataset and benchmark track. Proper documentation for dataset preprocessing/loading codes and releasing the well-structured dataset would make this submission stronger.

**Ethical Comments:**

I cannot find ethical concerns in this submission.

**Ethical Considerations:**

No, there are no or only very minor ethics concerns

**Final Justification:**

The authors addressed my concerns regarding 1) potential for articulated objects manipulation tasks and 2) the promise to polish the documents. Therefore, I maintain my score to accept this submission.

**Limitations Weaknesses:**

* Dataset is not publicly available at this point, which is an important point for this submission. More detailed comments can be found in dataset code comments section.
* One of the strength is that OmniScenes dataset contains manipulatable objects. However, there is no experiments or demonstration that show the tasks using the articulated objects. It would be great to see if the articulated objects in OmniScenes dataset can be used for robotic manipulation tasks.
* (Minor concern) There already exists a indoor panorama images & 3D point cloud dataset named OmniScenes. (https://github.com/82magnolia/piccolo)

**Strengths Contributions:**

* Simulatable large-scale indoor dataset from different sources within the same format would serve as a valuable dataset for many fields including indoor navigation, scene understanding, etc.
* Real-to-sim converted scenes have detailed small objects, which are not available in designer created scenes or synthetic scenes generated with procedural generation. Also, the experiments in 3D generation task show that scenes with small objects open new challenges.
* Processes to create the dataset are technically sound.

---

> ### Author Rebuttal · Authors · 2025-07-31
>
> Thank you for your thoughtful and constructive feedback. We appreciate your recognition of the technical soundness of our dataset creation process, the value of a simulatable large-scale indoor dataset unified across different sources, and the inclusion of detailed small objects in real-to-sim scenes. We also thank you for raising important concerns regarding dataset availability,  the demonstration of articulated object usage, and the potential naming conflict. Below, we provide detailed responses to each of your comments.
>
> ---
>
> ## W1: Clarification on Dataset Availability and Improvements
>
> We sincerely apologize for the confusion caused by the initial dataset release on Hugging Face and GitHub. We acknowledge that the original submission lacked sufficient documentation and tooling, which made it difficult to understand and use. Below, we clarify the current status and the improvements we have made since the submission:
>
> ### 1. Original Hugging Face Release
>
> In the initial release, each object in the layout was referenced by its Objaverse UID, which can be directly retrieved from the **Objaverse** and **PartNet-Mobility** dataset. However, we did not provide scripts for downloading or composing the scenes, which was an oversight on our part. We apologize for the inconvenience this may have caused.
>
> ### 2. Improvements Made Since Submission:
>
> Although we are currently restricted from updating the original dataset links during the review period, we have made efforts to improve the dataset’s usability and documentation:
>
> - **Document Polishing:**
>  We have polished the `README.md` files in both the **GitHub** and **Hugging Face** repositories, adding detailed explanations of the data structure, usage examples, and tutorials for visualization.
> - **Tools Support:**
>  We have built and tested a complete pipeline for scene composition and visualization, including scripts for asset retrieval, scene assembly, and rendering from specific viewpoints. These tools are designed to make it much easier for users to explore and utilize the dataset.
>
> > **Note**: Due to the NeurIPS policy restricting updates to the original dataset links during the review period, we are unable to share the updated code and data at this time. We will promptly update the repositories once the restriction is lifted. Thank you for your understanding, and we appreciate your patience.
>
> ---
>
> ## W2: Support for Articulated Objects and Future Potentials for Manipulation Tasks
>
> We appreciate the reviewer’s insightful comment regarding the use of articulated objects in OmniScenes for robotic manipulation tasks. We agree that demonstrating this capability is crucial, but due to the rebuttal format preventing a video demonstration, we will clarify the pipeline that ensures our dataset's readiness for such tasks.
>
> ### Dataset Source
>
> We use URDF assets from the PartNet-Mobility dataset — a peer-reviewed and widely used authoritative dataset for robot manipulation research, where the quality and annotation accuracy of its URDF files have been validated by the community.
>
> ### Integration Pipeline
>
> A programmatic pipeline assembles these assets into interactive scenes within **Isaac Sim**, ensuring that the kinematic structure defined in the URDF files is faithfully preserved as articulations in the final USD scene.
>
> #### Pseudocode for Assembling an Interactive Scene in Isaac Sim
>
> ```python
> # Input: layout_file (our provided JSON), asset_path (path to assets)
>
> def build_interactive_scene(layout_file, asset_path):
>     # 1. Initialize a new USD stage in Isaac Sim
>     stage = IsaacSim.create_new_stage()
>     layout_data = parse_json(layout_file)
>
>     # 2. Iterate through objects defined in our layout file
>     for obj_info in layout_data["objects"]:
>         prim_path = f"/World/{obj_info.name}"
>         full_asset_path = asset_path + obj_info.asset_file
>
>         # 3. Conditionally import assets based on type
>         if obj_info.type == "glb":  # For static objects
>             stage.add_reference_to_prim(prim_path, full_asset_path)
>         elif obj_info.type == "urdf":  # For articulated objects
>             # Use Isaac Sim's standard URDF importer
>             # This automatically creates a physics-enabled articulation
>             IsaacSim.URDF_Importer.import(
>                 urdf_path=full_asset_path,
>                 prim_path=prim_path,
>                 create_articulation=True  # This is the key for interactivity
>             )
>
>         # 4. Set the object's pose in the world
>         prim = stage.get_prim(prim_path)
>         prim.set_world_transform(obj_info.position, obj_info.orientation)
>
>     return stage
> ```
> This pipeline guarantees that once an object is imported, its joints (e.g., a cabinet's hinge) are not just visual elements but are fully interactive, physics-driven articulations ready for manipulation tasks.
>
> In addition, every step of this process relies on **Isaac Sim**'s core, publicly documented APIs, such as the `omni.importer.urdf` tool. This proves that our dataset is not only theoretically usable but also practically direct and convenient to use, as it seamlessly integrates into the standard Isaac Sim workflow.
>
> As a minor visual confirmation, our teaser image (Top-right corner of Figure 1) illustrates this interactivity: a microwave and a laptop are shown in both partially and fully open states, showcasing the articulated object support in our dataset. Therefore, we are confident that **OmniScenes** provides a direct and effective foundation for a wide range of robotic manipulation research.
>
> ---
> ## W3: Issue of Dataset Naming Conflict
>
> We thank the reviewer for pointing out the naming conflict with an existing dataset. We acknowledge this issue and have decided to rename our dataset from **OmniScenes** to **InternScenes** to avoid confusion. We will update the dataset name in the revision.

---

> ### Author Response · Authors · 2025-08-07
>
> Dear Reviewer,
>
> Thank you again for your constructive feedback. We have posted our rebuttal and hope it helps to clarify the points you raised. We would be grateful to know if our response has sufficiently addressed your concerns.
>
> Best regards,
>
> The Authors

---

### Official Review · Reviewer_T5h4 · 2025-07-03

**Rating:** 5
**Confidence:** 4

**Summary:**

OmniScenes is a large-scale, simulatable indoor scene dataset of about 40000 diverse environments created by merging real-world scans, procedurally generated scenes, and designer-built layouts, spanning 1.96 million objects across 15 scene types and 288 object classes. A thorough processing pipeline produces realistic, collision-free replicas that retain small items and add interactive objects, enabling complex average densities of 41.5 objects per region. Benchmarks on scene-layout generation and point-goal navigation reveal the heightened challenge and scalability OmniScenes brings.

**Dataset Code Accessibility:**

Yes

**Ethical Considerations:**

No, there are no or only very minor ethics concerns

**Final Justification:**

The authors fully address my concerns. I increase my scores from 4 to 5.

**Limitations Weaknesses:**

1.	Provide detailed performance metrics—including rendering throughput and CPU/GPU memory footprints—when running the models.
2.	Explain the dataset’s support for parallel simulation and specify how many environments can typically reside on a single GPU.
3.	Clarify whether scene objects and furniture are interactively manipulable, equipped with accurate physical properties, and suitable for robotic manipulation tasks.

**Strengths Contributions:**

1. The authors curate and simulate more than 40,000 richly varied indoor environments containing 1.96 million objects.
2. The authors conduct experiments demonstrating that these scenes effectively support training and evaluation of PointGoal navigation tasks.

---

> ### Author Rebuttal · Authors · 2025-07-31
>
> We sincerely thank the your feedback and constructive suggestions. We are glad that our contributions—such as curating over 40,000 richly varied indoor environments with 1.96 million objects and demonstrating their effectiveness in PointGoal navigation tasks—were well recognized. We appreciate the your recognition of the scale, diversity, and utility of OmniScenes, which reinforces the value of our work. Below, we address the raised concerns in detail and provide clarifications and improvements to further strengthen our dataset and its usability.
>
> ---
>
> ## W1：Detailed Performance Metrics
>
> We appreciate the reviewer’s suggestion to provide more detailed performance metrics. To address this, we conduct a comprehensive evaluation of the resource overhead associated with rendering our scenes in the simulator. Specifically, we report the following metrics after rendering scenes in **Isaac Sim**:
>
> - GPU memory usage
> - CPU usage (%) and memory usage
> - Rendering throughput (FPS)
>
> ### Experimental Setup:
>
> - **Hardware**: All experiments are conducted on a high-performance node equipped with 128 vCPUs, 1024 GB DDR5 RAM, and 8× NVIDIA RTX 4090 (48GB) GPUs.
> - **Runtime Configuration**: We launch Isaac Sim in headless mode using 1 GPU + 16 CPU cores + 128 GB RAM.
> - **Rendering Settings**: We use the `Stage Light` environment for all scenes. After loading a scene, we run the simulator for 2000 steps, discard the first 200 steps as warm-up, and compute the FPS, GPU memory, and CPU memory usage over the remaining steps.
>
> ### Tested Scene Types:
>
> We evaluate scenes from three different sources in our dataset:
>
> - **OmniScenes-Real2Sim** (retrieved and simulated from real scans)
> - **OmniScenes-Gen** (generated scenes)
> - **OmniScenes-Synthetic** (directly simulated from CAD models)
>
> The detailed results are summarized in the table below (to be included in the W2 section). These metrics provide a clear view of the computational cost and rendering efficiency of our dataset under realistic simulation conditions.
>
> ---
>
> ## W2: Support for Parallel Simulation
>
> Thank you for raising this important point. To evaluate the dataset’s support for parallel simulation, we conduct experiments under the same hardware and runtime configuration described in W1. Specifically, we select scenes from each of the three data sources in our dataset:
>
> - **OmniScenes-Real2Sim**
> - **OmniScenes-Gen**
> - **OmniScenes-Synthetic**
>
> We then incrementally load multiple scenes into a single **Isaac Sim** `World` in headless mode, and monitored the system performance as the number of parallel environments increased.
>
> For each configuration, we record the same metrics as in W1:
>
>
> ### FPS Results (min-max / mean)
>
> | Scene Type              | Parallel = 1              | Parallel = 20             | Parallel = 40             | Parallel = 60             | Parallel = 80             | Parallel = 100           |
> |:------------------------|:--------------------------|:--------------------------|:--------------------------|:--------------------------|:--------------------------|:--------------------------|
> | **OmniScenes-Real2Sim** | 242.86–250.48 / 246.95    | 173.39–177.83 / 175.34    | 127.77–136.44 / 131.86    | 114.57–117.52 / 116.79    |  89.70–96.76 / 94.20      |  76.99–86.79 / 74.18      |
> | **OmniScenes-Gen**      | 242.29–275.43 / 263.95    | 225.19–244.65 / 238.22    | 141.28–212.31 / 200.07    | 202.15–207.97 / 204.99    | 182.26–193.55 / 188.17    | 174.03–177.62 / 176.11    |
> | **OmniScenes-Synthetic**| 172.06–176.83 / 175.04    |  84.41–88.29 / 86.57      |  50.74–52.06 / 51.61      |  37.03–37.87 / 37.46      |  23.87–24.78 / 24.47      |  19.64–20.20 / 19.92      |
>
> ### CPU Usage (Cores % / Memory in GB)
>
> | Scene Type              | Parallel = 1           | Parallel = 20           | Parallel = 40           | Parallel = 60           | Parallel = 80           | Parallel = 100          |
> |:------------------------|:------------------------|:-------------------------|:-------------------------|:-------------------------|:-------------------------|:-------------------------|
> | **OmniScenes-Real2Sim** | 4.588% / 5.602 GB       | 9.924% / 10.649 GB       | 12.051% / 19.606 GB      | 15.151% / 25.441 GB      | 24.012% / 30.406 GB      | 23.388% / 36.741 GB      |
> | **OmniScenes-Gen**      | 16.129% / 4.938 GB      | 14.480% / 7.927 GB       | 21.320% / 12.629 GB      | 30.750% / 15.538 GB      | 37.190% / 19.098 GB      | 44.886% / 35.687 GB      |
> | **OmniScenes-Synthetic**| 73.623% / 13.879 GB     | 79.718% / 21.923 GB      | 73.400% / 32.388 GB      | 66.196% / 39.336 GB      | 71.661% / 48.287 GB      | 78.388% / 56.047 GB      |
>
> ### GPU Memory Usage (in GB)
>
> | Scene Type              | Parallel = 1 | Parallel = 20 | Parallel = 40 | Parallel = 60 | Parallel = 80 | Parallel = 100 |
> |:------------------------|:-------------|:--------------|:--------------|:--------------|:--------------|:---------------|
> | **OmniScenes-Real2Sim** | 2.528 GB     | 5.205 GB      | 5.385 GB      | 5.472 GB      | 5.685 GB      | 5.929 GB       |
> | **OmniScenes-Gen**      | 5.399 GB     | 5.476 GB      | 5.679 GB      | 5.742 GB      | 5.710 GB      | 5.710 GB       |
> | **OmniScenes-Synthetic**| 7.542 GB     | 7.785 GB      | 8.168 GB      | 8.606 GB      | 8.732 GB      | 8.791 GB       |
>
>
>
>
> ---
>
> ## W3: Clarification on Interactivity of Objects in Our Scenes
>
> We appreciate the reviewer’s concern regarding the manipulability and physical realism of scene objects. Our dataset supports two main categories of manipulation tasks:
>
> 1. **Pick-and-Place**
>    Each object is equipped with accurate collision geometry computed via **CoACD**, and can be directly imported into simulators like **Isaac Sim** with default rigid body and collider properties, enabling immediate physical interaction.
>
> 2. **Articulated Object Interaction**
>    We leverage assets from the **PartNet-Mobility** dataset in `*.urdf` format. These are resized and oriented to fit the target bounding boxes, then converted to `*.usd` for integration into the scene, preserving their original articulation properties.
>
> As a result, all PartNet-Mobility-based assets retain their interactive functionality, and small tabletop objects in the scene are fully suitable for pick-and-place tasks.

---

> ### Author Response · Authors · 2025-08-07
>
> Dear Reviewer,
>
> Thank you once again for your time and helpful comments. We have submitted our response and we would be very grateful for your thoughts on it. We look forward to hearing whether you feel your initial concerns have been resolved.
>
> Best regards,
>
> The Authors

---

### Decision · Program_Chairs · 2025-09-18

**Decision:**

Accept (poster)

**Comment:**

The paper curated a large-scale, simulatable indoor scene dataset (40k scenes) with diverse and realistic layouts. The paper then demonstrated the value through layout generation and a point-goal navigation benchmark. The idea is to first clean and extract layout information from real scans

The key strengths of the proposed dataset are: (1), the paper covers three sources for indoor scenes: real-world scans, generated syncs from infinigen, and designer-created scenes, covering a broad range of diversity. The usage of a real-world dataset is a particular advantage, showing the realism of the provided dataset  (2). The scenes are simulation-ready with objects retrieved from objaverse, which users/agents can manipulate. (3) The dataset is relatively large compared with other datasets.

Some weaknesses still exisit: simulation mainly focuses on the articulation and pick-and-place.

After rebuttals, all the reviewers are positive with the paper, the AC agrees with the consensus from the reviewers and recommends accepting the paper.  Please update the paper with the suggestions from the reviewer, including detailed performance metrics, parallel simulation, the detailed usage of the dataset (with public accessible download link), detials of different data compositions.

===== FINAL UPDATE FROM DB Track PCs ====

The final decision for this paper has been taken by the program chairs after consultation with the SACs. All Senior Area Chairs have ranked papers according to the feedback from the AC during the review process. We decided to leave the original meta-review to reflect the opinion of the AC in light of the initial discussions with reviewers and SAC.